# Dynamics and maintenance of categorical responses in primary auditory cortex during task engagement

Rupesh K Chillale[1,2], Shihab Shamma[1,3], Srdjan Ostojic[2], Yves Boubenec[1]*

[1]Laboratoire des Systèmes Perceptifs, Département d'Études Cognitives, École Normale Supérieure, PSL University,, Paris, France; [2]Laboratoire de Neurosciences Cognitives Computationnelle (INSERM U960), Département d'Études Cognitives, École Normale Supérieure, Paris, France; [3]Institute for System Research, Department of Electrical and Computer Engineering, University of Maryland, College Park, College Park, Maryland, United States

**Abstract** Grouping sets of sounds into relevant categories is an important cognitive ability that enables the association of stimuli with appropriate goal-directed behavioral responses. In perceptual tasks, the primary auditory cortex (A1) assumes a prominent role by concurrently encoding both sound sensory features and task-related variables. Here, we sought to explore the role of A1 in the initiation of sound categorization, shedding light on its involvement in this cognitive process. We trained ferrets to discriminate click trains of different rates in a Go/No-Go delayed categorization task and recorded neural activity during both active behavior and passive exposure to the same sounds. Purely categorical response components were extracted and analyzed separately from sensory responses to reveal their contributions to the overall population response throughout the trials. We found that categorical activity emerged during sound presentation in the population average and was present in both active behavioral and passive states. However, upon task engagement, categorical responses to the No-Go category became suppressed in the population code, leading to an asymmetrical representation of the Go stimuli relative to the No-Go sounds and pre-stimulus baseline. The population code underwent an abrupt change at stimulus offset, with sustained responses after the Go sounds during the delay period. Notably, the categorical responses observed during the stimulus period exhibited a significant correlation with those extracted from the delay epoch, suggesting an early involvement of A1 in stimulus categorization.

*For correspondence:
boubenec@ens.fr

## Editor's evaluation

This study provides an important contribution to our understanding of the neural basis for the categorical perception of sounds. Although the number of animals included is small, solid evidence is presented to show how categorical information emerges in the ferret primary auditory cortex following sound presentation and persists until a behavioral response is made. The work will be of interest to neuroscientists interested in the neural representation of task–related variables in sensory cortex during decision–making tasks.

## Introduction

Grouping individual stimuli into abstract, context-dependent categories and maintaining them in memory are fundamental cognitive abilities for appropriate behavioral responses and decisions. Neural correlates of such categorical perception have been found widely across cortical areas and

modalities, from sensory cortices to the frontal regions (*Swaminathan and Freedman, 2012*; *Folstein et al., 2013*; *Yin et al., 2020*; *Reinert et al., 2021*). The emerging picture thus far has been that categorization is a hierarchical process implemented through a series of computations and gradual transformations across multiple cortical areas, beginning with relatively basic stimulus representations in the primary sensory areas, and ultimately concluding with categorical responses in the frontal regions (*Yin et al., 2020*; *Atiani et al., 2014*; *Tajima et al., 2017*). In the auditory modality, this framework suggests that the primary auditory cortex (A1) would mainly represent stimulus acoustic features and only exhibit weak correlates of the associated behavioral categories (*Jaramillo et al., 2014*; *Selezneva et al., 2017*; *Christison-Lagay and Cohen, 2018*).

Recent studies have however challenged this conception from multiple viewpoints. To begin with, the encoding of sounds in the primary auditory cortex (A1) is relatively plastic and is rapidly enhanced by task engagement (*Yin et al., 2020*; *Fritz et al., 2003*; *Fritz et al., 2007*; *Xin et al., 2019*). Furthermore, A1 has already been shown to extract some of the behavioral meaning of target sounds for task-relevant downstream readout (*Rodgers and DeWeese, 2014*; *Bagur et al., 2018*; *Barbosa et al., 2022*). Nevertheless, the stimuli and experimental designs used in most of these experiments did not allow for disentangling category-specific neural dimensions from sensory-related activity. It thus remains uncertain how A1 population responses represent stimulus category during sound presentation in addition to strictly sensory-evoked responses, and how the representations of the task-relevant categories could be dynamically maintained after the stimulus is played. This study explores the critical questions of how and when categorical encoding emerges in primary auditory cortex, and the extent to which these categorical representations become behaviorally shaped upon task engagement.

To this end, we trained ferrets to classify click trains into Go and No-Go categories of either low or high rates during an appetitive Go/No-Go task. After training, we recorded in A1 while ferrets passively listened to or actively discriminated between the two stimulus categories. Using population-level analyses, we contrasted the representation of stimulus features (click-rates) and behavioral categories (Go/No-Go) during and after stimulus presentation. First, we discovered a signature of category which emerged early during stimulus presentation in the population average. Second, upon task engagement, the population-level representations of the No-Go category became suppressed, leading to an asymmetrical representation of the Go stimuli relative to the No-Go sounds and pre-stimulus baseline. Third, at stimulus offset, the population code changed abruptly and a large fraction of neurons maintained sustained responses after Go sounds throughout the delay epoch. The responses to Go sounds built up during the delay in anticipation of the subsequent behavioral response. Lastly, incorrect behavioral choices could be traced back to degraded sensory encoding during the stimulus period, resulting in a degraded categorical representation.

## Results

### A1 neurons sustain activity after Go sounds during task engagement

Two ferrets were trained on a Go/No-Go delayed categorization task under appetitive reinforcement. Water-deprived ferrets had to classify click trains into two categories: target (Go) and non-target (No-Go) depending on the rates of click trains. Six rates were used, from 4 to 24 Hz in 4 Hz steps, and with a category boundary fixed at 14 Hz. To ensure the dissociation between categories and stimulus rates, one animal was trained with low rates as the Go sounds, while the second animal classified high rates as the Go sounds. Click trains were presented for 1.1 s and were followed by a 1-s-long delay in which the animals had to refrain from licking (*Figure 1a*). Licks during the subsequent 1-s-long response window were rewarded with water in Go trials and punished with a timeout in No-Go trials. Any licks before this response window (early licks) resulted in an aborted trial and were punished with a timeout; these were called 'early trials'.

Ferrets categorized the click trains (ferret P: d' = 1.2 ± 0.5, n = 35 sessions; ferret T: d' = 1.1 ± 0.2, n = 39 sessions; *Figure 1b*) with a bias to lick for the No-Go rates close to the category boundary (12 Hz for the animal shown in *Figure 1b*). This was found in both ferrets (*Figure 1—figure supplement 1a and b*) regardless of the mapping between stimulus rates and category, suggesting that their decision criterion was relatively liberal and impulsive. First lick probability was higher in Go than in No-Go trials throughout the entire trial duration (*Figure 1c*), confirming the categorical response profile. We also

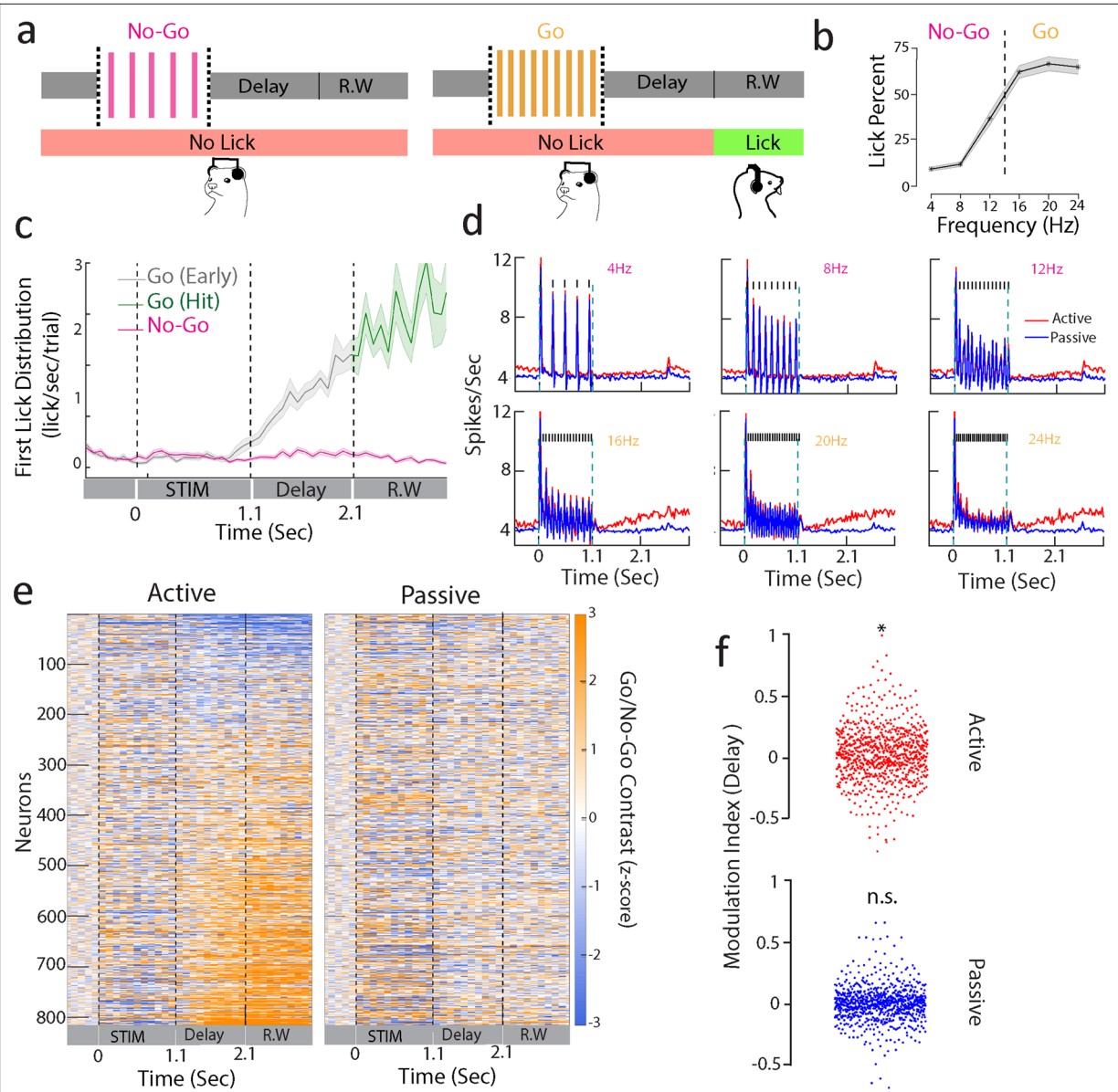

**Figure 1.** Sustained A1 activity during a delayed categorization task. (**a**) Delayed categorization task. A trial starts with a 0.5 s of pre-stimulus silence followed by 1.1-s-duration click train stimulus. The animal must wait for a 1-s delay period before the 1-s-long response window (R.W.). Correct trials were rewarded with water while error trials and early trials (lick during delay period including sound period) were punished by a timeout. (**b**) Proportion of licks in response window for one of the animals (ferret P) with low rates as No-Go and higher rates as Go stimuli. Only non-early trials are considered. Shaded regions are SEM. (**c**) Temporal profile of first lick rate. Shaded regions are SEM. (**d**) Average PSTHs of all neurons from ferret P corresponding to passive (blue curve) and active (red curve) states for each of the click trains (only correct behavioral trials were used, i.e., correct rejections for No-Go and hits for Go sounds). Note that the response during the delay period is enhanced for Go stimuli (16, 20, 24 Hz). (**e**) Contrast (Go – No-Go firing rate) computed for neurons of both animals (n = 816 units) in passive (right) and active (left) states (z-scored with pre-stimulus baseline activity). Neurons were ranked by delay firing rate for Go active trials. (**f**) Modulation index (see 'Methods') with respect to spontaneous activity during delay (1.1–2.1 s) period for active and passive states (*t*-test *p<0.05; n = 816 units from both ferrets).

The online version of this article includes the following figure supplement(s) for figure 1:

**Figure supplement 1.** Behavioral performance and sustained responses for all animals.

**Figure supplement 2.** Modulation index for delay activity in (**a**) ferret P and (**b**) ferret T (*** p<0.001, * p<0.05).

**Figure supplement 3.** Difference in average firing rate between Go and No-Go correct trials computed for neurons of both animals (n = 816 units) in active (left) and passive (right) states (z-scored with pre-stimulus baseline activity).

observed a build-up of lick probability during the delay, with early responses being quite common in Go trials. Early trials and error trials (misses and false alarms) were discarded in the analysis unless otherwise specified.

We chronically recorded neural activity from a total of 816 units in A1 (ferret P: 575; ferret T: 241), while the two animals alternately listened passively to the stimuli or actively engaged in categorizing them. *Figure 1d* depicts the average click responses across all cells, showing a significant increase in firing rate relative to baseline activity in all conditions. Firing rates also steadily ramped up during the delay period in Go trials compared to No-Go's (*Figure 1d and e*; both animals in *Figure 1—figure supplement 1c and d*). Importantly here we considered only hit trials in which the animals were not licking the spout until the response window. Neurons with significant changes in firing rate during the delay period did not show any modulation of activity during the passive state (*Figure 1e*). Ranking neurons by delay activity during the passive condition did not reveal a similar pattern neither in passive or active states (*Figure 1—figure supplement 3*), reflecting a pattern of response specific to the task-engaged delay period. Additionally, we found that the firing rates of these neurons increased during the delay (*Figure 1f*, paired *t*-test active p=0.01, passive p=0.15, n = 816 units; both animals in *Figure 1—figure supplement 2*; ferret P: active p=0.04, passive p=0.83, n = 575 units; ferret T: active p<0.001, passive p=0.35, n = 241 units). Nevertheless, we noticed some heterogeneity in the delay activity, with a proportion of neurons showing large suppression during this period (top neurons in *Figure 1e*).

## A1 population activity encodes behavioral categories during stimulus in both active and passive states

Because A1 activity was heterogeneously modulated during the post-Go sound delay (*Figure 1f*), we relied on population decoding to examine the categorical representation. To do so, we trained linear decoders to discriminate Go and No-Go trials based on single-trial population activity (*Figure 2a*). During task engagement, population responses steadily discriminated between Go and No-Go categories along the entire course of the trial (*Figure 2a*, both animals in *Figure 2—figure supplement 1*). Decoding accuracy diminished at stimulus offset, but subsequently increased during the delay period, ultimately peaking in the behavioral response window (*Figure 2a*, *Figure 2—figure supplement 1*). By contrast, accuracy in the passive context decreased throughout the delay period. As a control, we performed recordings in an untrained (naive) animal using the same set of stimuli (*Figure 2—figure supplement 2a*). The accuracy of passive context decoder was also similar to what is observed in the naive animal (*Figure 2—figure supplement 2b*). Note that decoding was performed at the single-session level (11.3 ± 4.9 neurons per session for ferret P; ± SD; n = 35 sessions and 5.2 ± 3.2 neurons per session for ferret T; n = 39 sessions), explaining the modest but above chance-level performance of the decoder.

We then looked further into how population code evolves during the trial. For this, we compared decoders trained at different time bins throughout the trial. We found that for both passive and task-engaged sessions, the direction of the decoding axis was mainly preserved during the stimulus epoch (black outline in *Figure 2b*), and the corresponding decoders trained on passive and task-engaged data were correlated (*Figure 2—figure supplement 3*). Furthermore, these decoders were uncorrelated with the decoding axis during the *delay* period of the task-engaged condition (white outline in *Figure 2b*), indicating a behavior-dependent change of the population code at the sound offset. This delay activity pattern further persisted during the *response window*, as shown by the correlation between delay and response window decoding axes (pink outline in *Figure 2b*). This suggests a time-independent categorical representation in A1 population activity throughout the delay and response window epochs.

While categorical encoding is evident during task engagement *after* the stimulus end, it remains unclear whether it was biased toward stimulus-related or category-related responses *during* the stimulus. Indeed, both stimulus-related (click train rate) and category-related responses are captured by the decoders (*Figure 2—figure supplement 4a and b*). To disentangle sensory and categorical contributions, we computed a population-level linear regression of the neuronal firing rates at each time bin in each session (see 'Methods' for details). Single-neuron activity showed linear relationship with click rates (example in *Figure 2c*) or robust categorical encoding during and after the stimulus (*Figure 2d and e*). To model this type of responses, we used two regressors: (i) stimulus rates as a *sensory*

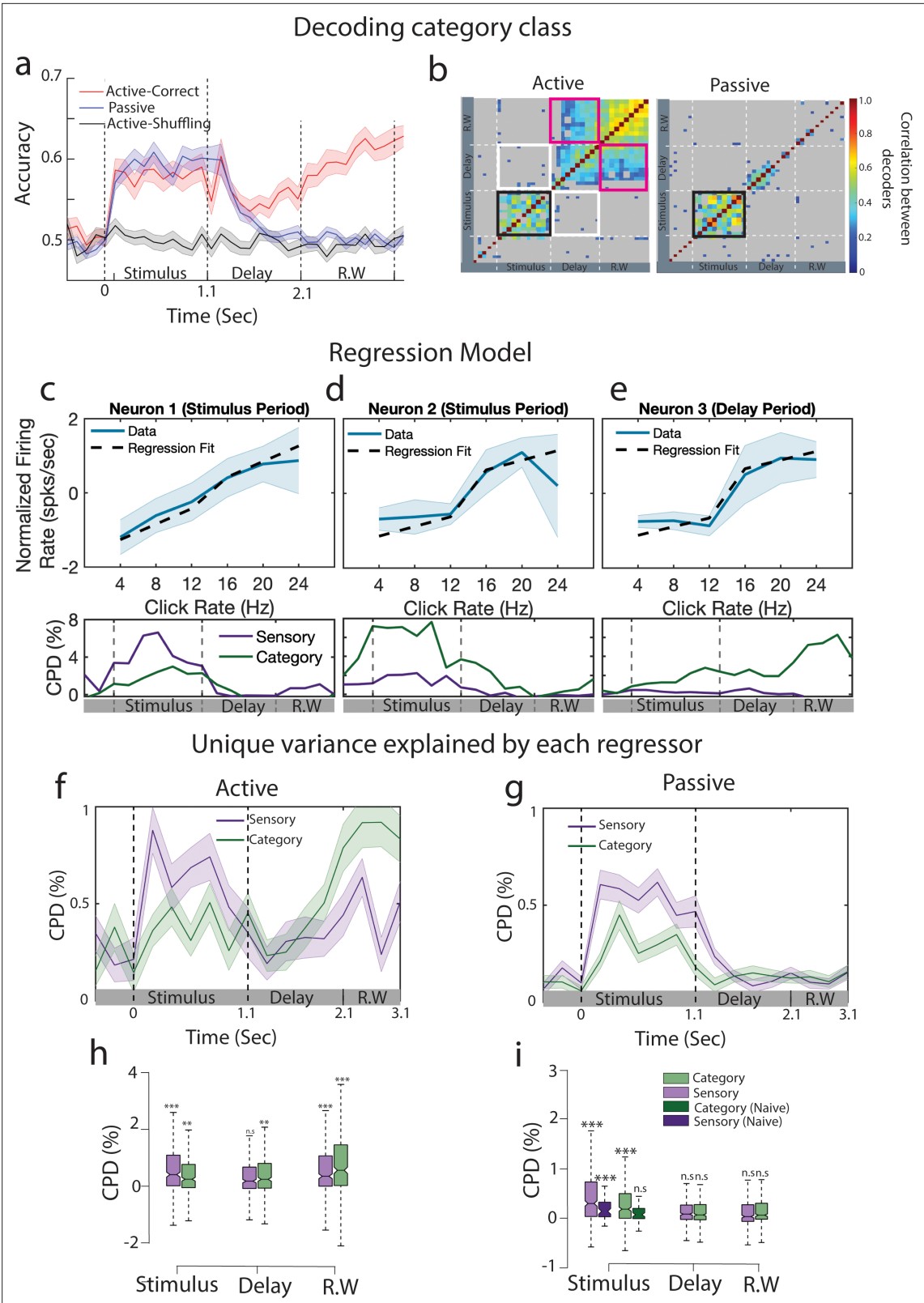

**Figure 2.** Emergence of categorical information from primary auditory cortex during stimulus presentation. (**a**) Go vs No-Go classification performance in active (red curve) and passive (blue curve) conditions (n = 35 sessions from ferret P). Gray curve indicates the performance by shuffling labels for the task-engaged condition. Error bars show 1 SD. The dashed lines separate stimulus, delay, and response window (R.W.) periods. (**b**) Cross-correlation matrix between decoders trained at different time points for task-engaged (left) and passive (right) data. Nonsignificant correlations are shown in

*Figure 2 continued on next page*

*Figure 2 continued*

gray. Significance was assessed by permutation test (200 permutations). Black frame shows the significant cross-correlation during the stimulus period. White frames show absence of correlation between decoders trained during the stimulus and the delay period. Similarly, pink frames correspond to correlation of stimulus and response window decoders. (**c**) Example neuron showing sensory encoding during sound period, firing rate plotted as a function of click rates (blue curve, error bars show 1 SD over trials). Prediction from the regression model are overlaid (dashed line) ($CPD_{sensory,stimulus}$ = 0.07 ± 0.02/$CPD_{category,stimulus}$ = 0.02 ± 0.01). Time course of coefficient of partial determination (CPD) is shown below. (**d, e**) Same as (**c**) for example neurons mostly tuned to categories during the stimulus (**d**, $CPD_{sensory,stimulus}$ = 0.02 ± 0.01/$CPD_{category,stimulus}$ = 0.05 ± 0.01) or delay (**e**, $CPD_{sensory,delay}$ = 0.00 ± 0.01/$CPD_{category,delay}$ = 0.02 ± 0.01) periods. (**f**) CPD computed by fitting linear regression models in active state (n = 395 neurons from ferret P). Shaded region represents 1 SEM over all the neurons. (**g**) Same as (**f**) for passive state. (**h**) CPD computed during the stimulus, delay, and response window time epochs. Significance is tested against pre-stimulus period value (two-tailed *t*-test ***p<0.001, n = 395 neurons from ferret P). (**i**) Same as (**h**) for passive state. CPD computed from a naive animal is added in the stimulus period for comparison.

The online version of this article includes the following figure supplement(s) for figure 2:

**Figure supplement 1.** (**a,c**) Decoding accuracies for passive (blue) and active (red) states for ferret P (n=35 sessions). (**b,d**) Decoding accuracies for passive (blue) and active (red) for ferret T (n=39 sessions).

**Figure supplement 2.** Decoding accuracy in naive animals.

**Figure supplement 3.** Correlation of passive and task-engaged decoders for ferrets P and T.

**Figure supplement 4.** Projection of each click rates onto linear decoders.

**Figure supplement 5.** Regression model supplemented with licks and reward.

regressor and (ii) the binary category (Go or No-Go) as a *category* regressor. We also fitted two additional models in which the labels of one of the regressors were shuffled (*Musall et al., 2019*), which allowed us to compute the relative contributions using a *coefficient of partial determination* (CPD) from each regressor as the decrement in variance due to shuffling one of the regressors (*Fisher et al., 2019*). During task engagement, we found that the CPD of the category regressor increased during the stimulus period and persisted throughout the delay period, before increasing at the response window (*Figure 2f and h*; paired *t*-test on category regressor, delay vs response window; ferret P: n = 395 neurons, p<0.001; ferret T: n = 203 neurons, p<0.01). Interestingly, both task-engaged and passive category CPD were larger than chance during stimulus presentations (*Figure 2f–i*), an effect absent in the naive animal (*Figure 2i*), which indicates training-dependent encoding of categories. Because the variance captured by the category regressor during the delay could be related to other factors than pure behavioral categories, such as reward delivery or licking, we fitted another model including lick and reward regressors (see 'Methods'). We found similar category axes in the passive and active conditions when accounting for licks and rewards (*Figure 2—figure supplement 5*). We consider further the possible link of this delay activity pattern with reward expectation or motor preparation (see 'Discussion'). For practical reasons, we will refer to the delay activity as category-related in the following sections.

## Population-level categorical responses show suppression of No-Go representations upon task engagement

We then examined how the category information during the delay emerged from the mixed representation of stimulus and category we found during stimulus presentation. Regression axes define encoding axes along which population activity carries information about stimulus click rate or category. We thus projected population activity onto the category neural axis to track the categorical representation in the population at each moment in time. This procedure sums the neuronal responses weighted by the coefficients extracted from the earlier regression analysis and reduces A1 population dynamics to one information-bearing dimension encoding category through time, independently from sensory information. We did so in both the stimulus and the delay periods for the passive and task-engaged conditions in order to compare the patterns of categorical encoding across conditions. Projections on the category neural axis (called thereafter categorical responses) revealed a stable encoding of categories during sound presentation, regardless of the stimulus identity (*Figure 3a and b*), indicating that these categorical responses were not contaminated by sensory activity.

Categorization can be defined as the maximization of neural differences between categories and a minimization of the differences within a category. Therefore, we investigated stimulus discriminability along the category neural axis. We designed a category index (see 'Methods') which compared the

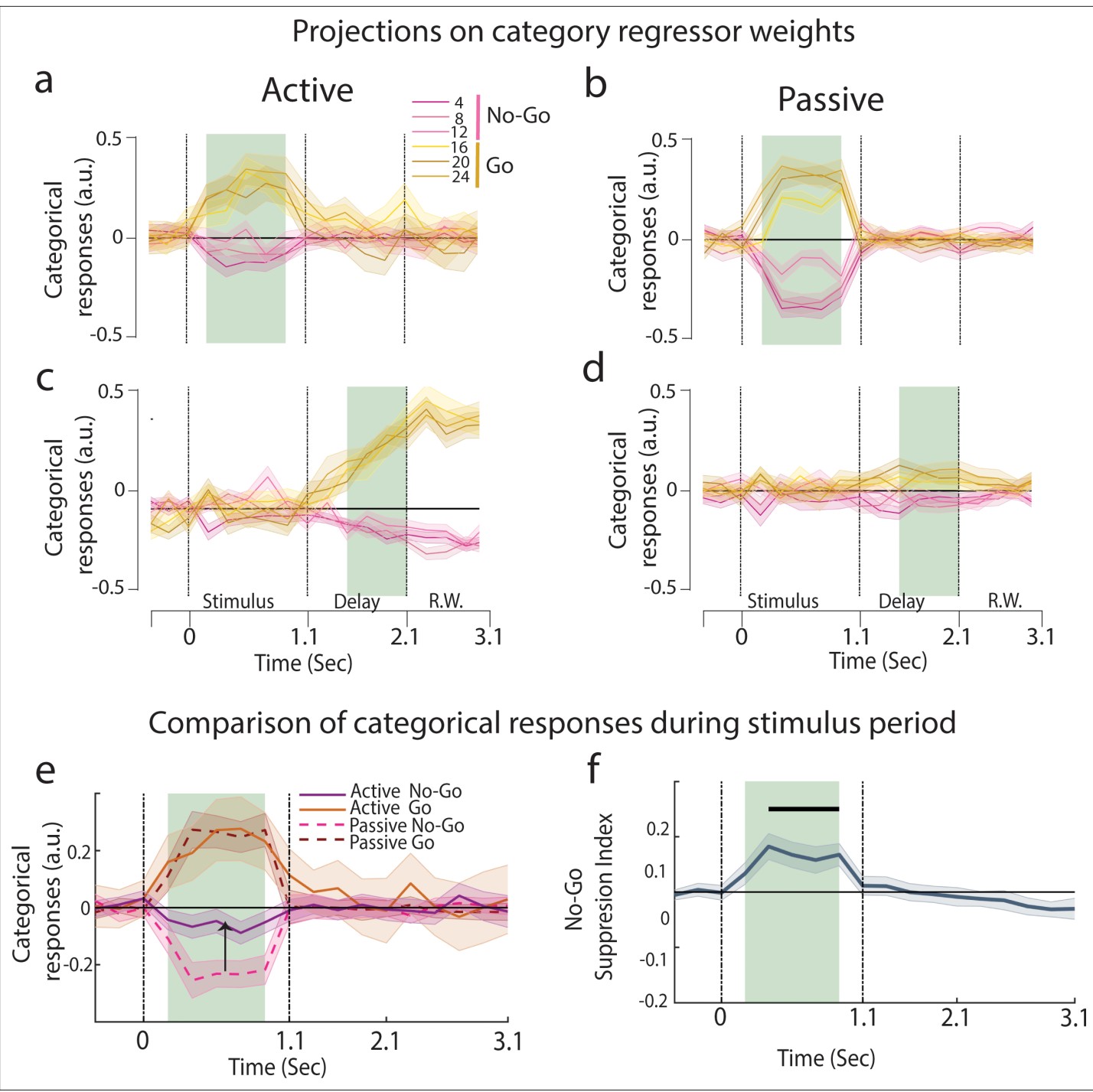

**Figure 3.** Emergence of categorical representation through suppression of No-Go sounds. (**a–d**) Projection of trial-averaged activities of individual click rates onto category axis trained at different time epochs (n = 35 sessions from ferret P). The shaded regions show training time and the error bars are ±2 SEM over sessions. (**a, c**) are active and (**b, d**) are passive states. See *Figure 3—figure supplement 2* for graded sensory responses to the different stimuli. (**e**) Categorical responses for passive and active states (n = 35 sessions from ferret P; *Figure 3—figure supplement 1d* for ferret T). (**f**) Time course of No-Go suppression in categorical responses (difference between the absolute value of passive and active No-Go categorical responses highlighted in **e**). Black bar represents the significant period (p<0.05, *t*-test n = 35 sessions).

The online version of this article includes the following figure supplement(s) for figure 3:

**Figure supplement 1.** No-Go suppression in all animals.

**Figure supplement 2.** Projection of trial-averaged activity onto sensory regressor weights.

**Figure supplement 3.** Projection onto category neural axis extracted from the task-engaged condition for passive and active states.

neural distance between stimuli at the category boundary (12 and 16 Hz) against pairs of stimuli within categories. We found that categories were effectively present in both passive and task-engaged states during stimulus presentation with equal magnitude (*Figure 3a and b*, *Figure 3—figure supplement 1a and b*). In contrast, significant categories were found only in the task-engaged state during the delay period (*Figure 3c and d*).

To further determine if task engagement induced a targeted change along the categorical neural dimension, we examined the temporal profiles of the Go and No-Go categorical responses. Here, we found that there was a clear shift of population coding between the two behavioral conditions: No-Go categorical responses aligned with the pre-stimulus spontaneous activity during task engagement (*Figure 3a*), inducing an asymmetry between No-Go and Go categorical responses (*Figure 3e*, both animals in *Figure 3—figure supplement 1c and d*), a characteristic that is not observed in the passive projections (*Figure 3b*). A No-Go suppression index measuring the relative displacement of the No-Go responses closer to the projections of spontaneous activity (*Figure 3f*) showed that the suppression was confined to the stimulus period (*Figure 3—figure supplement 1e and f*, bottom row; ferret P: p<10⁻⁶, n = 35 sessions; ferret T: p<10⁻⁴, n = 39 sessions), and absent in the delay period (ferret P: p=0.70; ferret T: p=0.46). Crucially, projections of the passive population responses onto the active category neural axis *did not* show the suppression of No-Go sounds, confirming that this shift originated from a change in the structure of the population responses during task engagement, and not due to the weights of the regressor themselves (*Figure 3—figure supplement 3*). In summary, these findings demonstrate a task-induced asymmetry in categorical representations which specifically targets the relevant neural dimension.

## Single-trial categorical representations correlate between sound and delay periods

The categorical signal in the stimulus period exhibits short latencies (*Figure 3a*) consistent with a feedforward mechanism of early categorization taking place in A1. We wondered whether this early categorical information may influence later categorical information captured in the delay period. However, the categorical axes in the early stimulus and delay periods were not correlated (white frames in *Figure 4a*). So how could the categorical information present in these two distinct intervals be still linked? To find out, we tested directly whether the categorical responses during the stimulus period correlated with those during the delay. Specifically, we examined the *single-trial* fluctuations of categorical responses in each period and discovered that the two categorical responses were correlated (one session in *Figure 4b*, all sessions in the inset; p<0.001, permutation test, n = 74 sessions). Note that the stimulus and delay categorical responses were computed from uncorrelated projection axes (*Figure 4a*, *Figure 4—figure supplement 1*) that were originally derived from distinct non-overlapping epochs, and hence the single-trial fluctuations in categorical responses need not have been correlated in any way.

## Post-sound anticipatory activity builds up towards behavioral response

Another interesting relationship is that between the categorical responses and the licks. We have observed that population activity on hit trials ramped up throughout the delay period (*Figures 3c and 4c*, categorical responses during the delay). By aligning single-trial projections onto lick timings during the response window, we found that the categorical activity built up during the delay period and culminated when the animal licked (*Figure 4d*, categorical responses centered on licks), indicating that delay activity was anticipating behavioral responses. We further tested whether the temporal dynamics of the categorical responses correlated with the timing of the animal's behavioral response. To do so, we took advantage of early Go trials when the animals licked *before* the response window (see lick histogram in *Figure 1c*), so that we accessed to Go trials in which first licks occurred earlier than what we had in hit trials. We then sorted all the collected trials into three groups: early trials with licks early in the delay period (first half), early trials with licks late in the delay (second half), and hits. We found significantly faster response build-up rates in early trials than in hit trials (*Figure 4c*). Early trials with early licks in the delay exhibited the steepest response build-ups (*Figure 4c*), a pattern that was not observed in the categorical responses during the stimulus period (*Figure 4—figure supplement 2*). We then aligned all trials to their respective reaction times and found that categorical responses of early and hit trials culminated at the licking time (*Figure 4d*). This alignment was done on

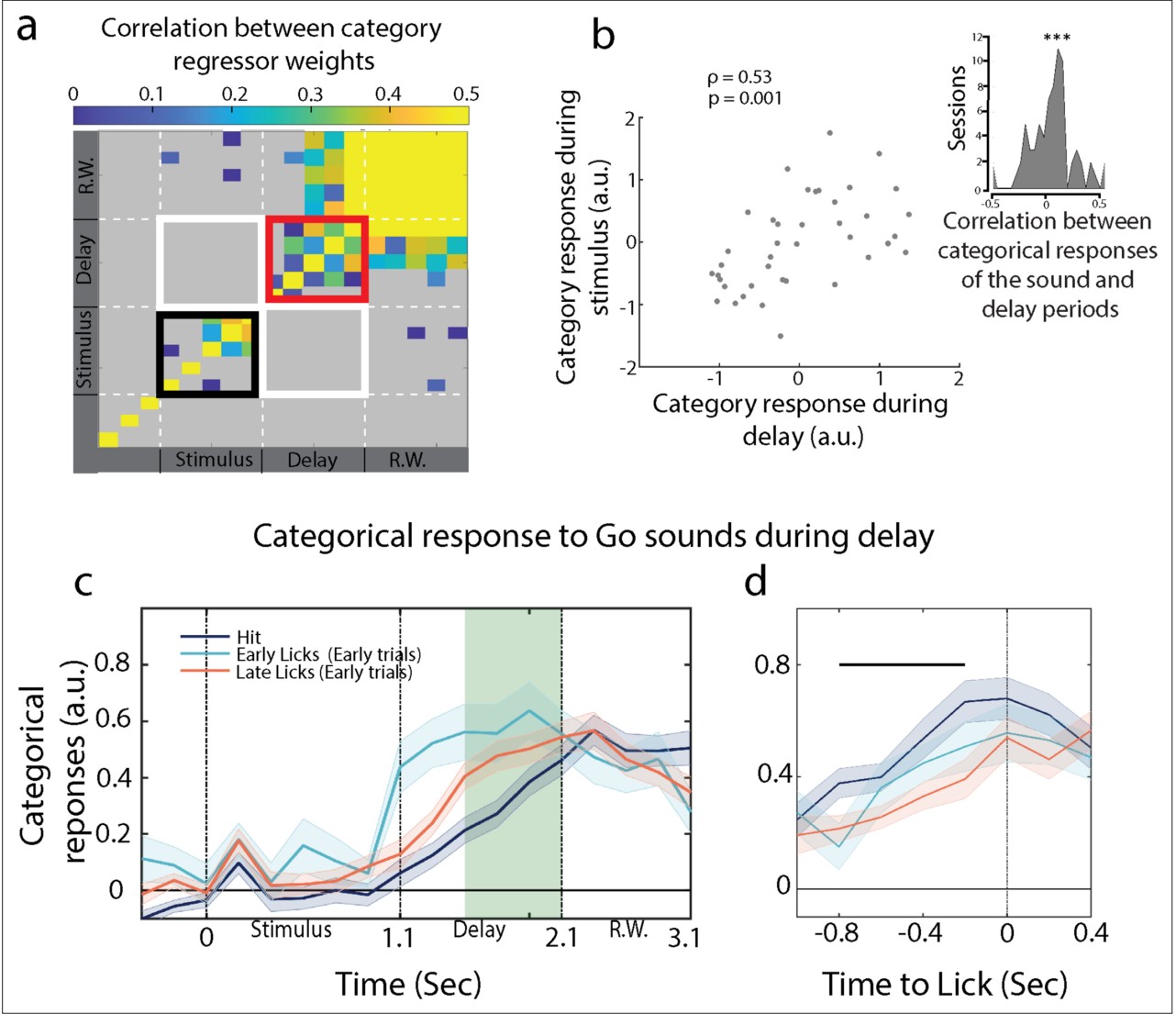

**Figure 4.** Post-stimulus anticipatory activity correlates with categorical representation during the sound. (**a**) Temporal evolution of the category axis (n = 35 sessions from ferret P). (**b**) Scatter plot for categorical response during stimulus and delay period for one session. Inset: correlation of single-trial categorical responses between the sound and delay periods (n = 74 sessions from both ferrets). (**c**) Categorical responses for hit, early trials with licks in the early (early licks) and late (late licks) phases of the delay period. (**d**) Lick-aligned categorical responses. Projections were not different between hit and early trials at lick time (see outcome decoding in *Figure 4—figure supplement 3*). The black bar represents the significant period (p<0.05, *t*-test).

The online version of this article includes the following figure supplement(s) for figure 4:

**Figure supplement 1.** Scatter plot of the category weights regressed during stimulus and delay periods (n = 395 neurons from ferret P and n = 203 neurons from ferret T).

**Figure supplement 2.** Projection of trial-averaged activity of hit, early–early, and late–early trials onto the category neural axis computed during the stimulus period.

**Figure supplement 3.** Delay activity in hit and early trials is identical.

lick times, that is, independently of neural activity, indicating a build-up of population-level dynamics anticipating behavioral response.

## Sensory information is degraded during error trials, leading to poor categorization

Finally, we searched for the neural correlates of incorrect perceptual decisions. In particular, we wanted to pinpoint which encoding stages were degraded during incorrect categorizations. We envisioned

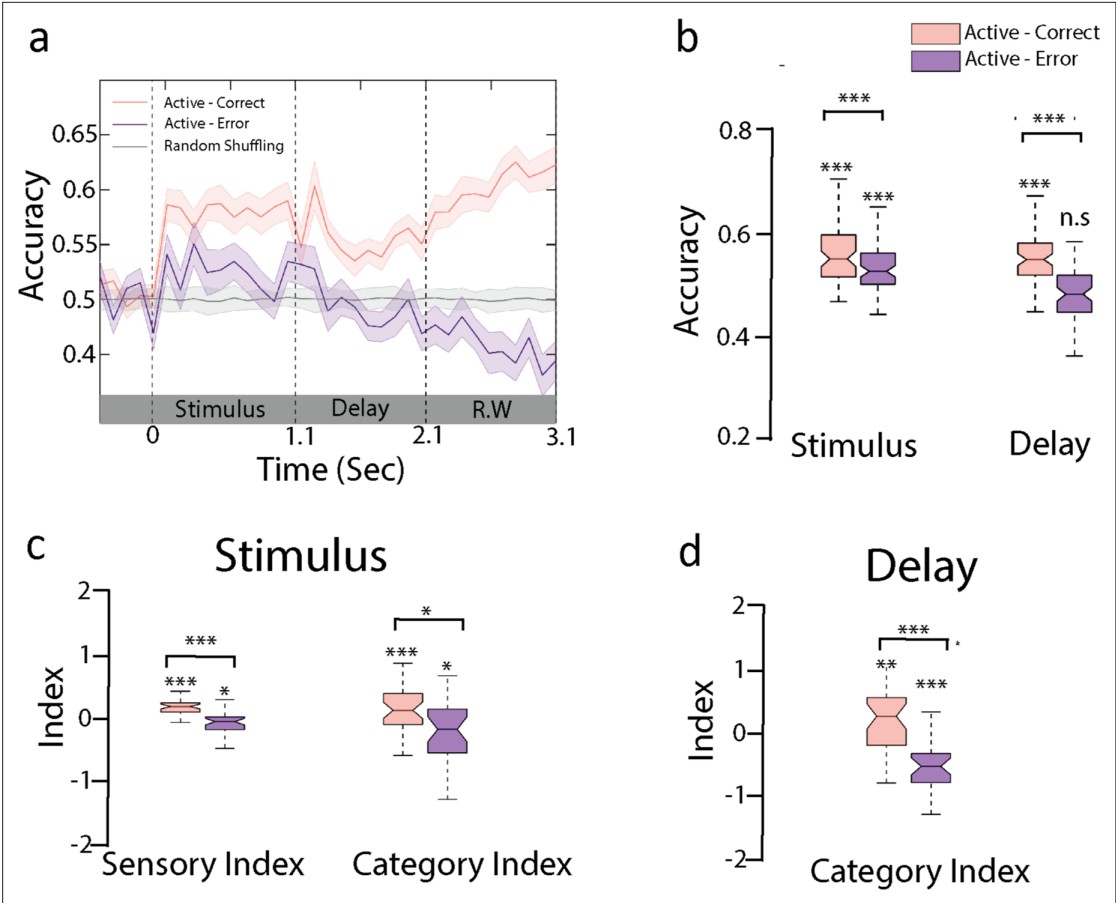

**Figure 5.** Sensory and categorical responses during error trials. (**a**) Encoding of behavioral choices: here we trained the active classifier on correct behavioral choices (hit and correct rejection) and used the decoding weights to compute the accuracy using incorrect behavioral choices (false alarm and miss) as shown in the purple curve (n = 35 sessions from ferret P). Gray curve indicated the performance by shuffling labels in task engagement. Error bars show 1 SD. (**b**) Accuracy of the active decoder for correct and error trials during sound and delay period (n = 35 sessions from ferret P). (**c**) Category index and sensory index computed from the projections of trial-averaged correct and error trials onto category axis trained during sound period (see 'Methods'). (**d**) Category index for delay period (see ***Figure 5—figure supplement 2*** for both animals).

The online version of this article includes the following figure supplement(s) for figure 5:

**Figure supplement 1.** Projection of trial-averaged activity of error trials (average number of error trials in ferret P: 55.1 ± 14.9 and in ferret T: 66.2 ± 13.2) in both Go and No-Go categories onto the classifiers trained on correct trials during stimulus presentation (**a**), delay (**b**), and response window (**c**).

**Figure supplement 2.** Sensory and Category index per animal.

three different hypotheses causing the error trials: (a) sensory information was intact in error trials, but conversion to the correct category during the stimulus was incorrect; or that (b) sensory-to-category processing during the stimulus period is correct, but the anticipatory activity is impaired during the delay; or that (c) sensory information was in fact degraded early on during the stimulus, leading to a loss of categorical information.

We first sought to test whether the error trials correlated with an inversion of the neural responses between the Go and No-Go categories, as would be predicted by hypothesis (a). To do so, we used Go/No-Go classifiers trained with correct trials (as in ***Figure 2a***) and reported the classifiers' performance in predicting the expected (and not actual) behavioral categories. If false alarms were fully behaving like hits, and misses like correct rejections, we should observe below-chance decoding accuracy. That was not the case, but instead we found a decrease in decoding accuracy during the sound period and a decoding performance at chance level during the delay (***Figure 5a and b***, ***Figure 5— figure supplement 1*** for separate projections of false alarm and miss trials; decoding accuracy during delay for incorrect trials: p=0.99, permutation test). As a control, we noted that decoding accuracy dipped below chance during the response window confirming that motor information was confined

to this late time period (*Figure 5a*). This indicates that incorrect decisions could be traced back to the stimulus period, excluding hypothesis (b) in which only the delay activity was incorrect during error trials.

We then further tested the hypothesis (c) of a degradation of sensory information during the stimulus period. In line with this interpretation, we found that sensory responses were lower in incorrect compared to correct trials (*Figure 5c*, *Figure 5—figure supplement 2a and b*; stimulus period-sensory index: ferret P: $p<10^{-5}$, n = 35 sessions; ferret T: $p<10^{-3}$, n = 39 sessions; stimulus period-category index: ferret P: p=0.01, n = 35 sessions; ferret T: p=0.35, n = 39 sessions; paired *t*-test across sessions, individual boxes were tested with permutation test). Categorical responses were further degraded during the delay period (delay-category index [*Figure 5d* and *Figure 5—figure supplement 2b and c*] for both animals; paired *t*-test across sessions; ferret P: $p<10^{-4}$, n = 35 sessions; ferret T: $p<10^{-5}$, n = 39 sessions). Altogether, this suggests that errors likely originated from an improper encoding of the stimulus that subsequently led to an incorrect categorization and behavioral response.

## Discussion

Categorical perception of real-world signals is a key cognitive function in all sensory modalities, one that is thought to be implemented at higher cortical levels (*Swaminathan and Freedman, 2012*; *Folstein et al., 2013*; *Yin et al., 2020*; *Freedman and Assad, 2006*; *Roy et al., 2010*; *Bizley and Cohen, 2013*). In this work, we examined if and how population responses in *primary* auditory cortex could contribute to and reflect the categorical encoding of sounds during passive listening or engagement in a Go/No-Go categorization task. The study resulted in four main findings. First, we isolated an encoding of behavioral categories in the population-average during stimulus presentation in both task-engaged and passive listening. This was not observed in a naive animal (*Figure 2i*). We interpret this observation as an effect of long-term memory (*Elgueda et al., 2019*). Second, we found that population-level categorical responses to No-Go sounds were suppressed during task engagement during the stimulus period (*Figure 3*). Third, the population-level categorical representation changed at stimulus offset but still correlated at the single-trial level with categorical responses observed during the stimulus period (*Figures 3 and 4*). Fourth, incorrect behavioral choices were traced back to degraded categorical encoding during the stimulus period, resulting in a degraded categorical representation during the subsequent *delay* period (*Figure 5*). Altogether, these results suggest that behavior-dependent categorical information emerged during the stimulus period, influenced population dynamics beyond the stimulus period itself, and persisted throughout the delay period until the behavioral response.

### Neuronal selectivity to behavioral categories during task engagement

Previous findings have found categorical responses in A1 with a marked increase of response contrast at category boundaries (*Yin et al., 2020*; *Xin et al., 2019*). Here, we focused on revealing the temporal dynamics and evolution of categorical encoding both in the population average and in the population code. Regression analyses allowed us to disentangle the binary categorical representation from the more graded representation of the sensory properties (i.e., click train rates) (*Figure 3—figure supplement 2*). Similar to a previous study in the mouse (*Xin et al., 2019*), we found that the encoding of stimulus sensory properties within behavioral categories was stable across behavioral states. In *Xin et al., 2019*, the authors found that categorical representations in A1 matched behavioral decision as early as at stimulus offset, and was maintained during several seconds. Such choice-related activity has been demonstrated to be the result of feedforward and feedback flow between primary sensory cortices and downstream regions (*Dehaene and Changeux, 2011*; *Yang et al., 2016*). Our observations are consistent with this view as we found that perceptual choice influenced the category decoded during the response window (*Figure 5a*). Nevertheless, we used longer stimulus followed by a long delay in our study, which allowed us to decouple the early categorical information building up during stimulus presentation from the late choice report of the animal. In addition, categorical information declined during stimulus and delay periods, suggesting that incorrect trials were due to a lack of proper categorical information early on during the stimulus itself.

## Population-level suppression of No-Go categorical responses

Our analyses showed that, even though categorical encoding was present in passive and active states in trained animals, task engagement induced a suppression of No-Go responses during the stimulus. This was not due to a lack of sound-evoked responses to No-Go sounds, as shown by the average population peristimulus time histogram (PSTH) (*Figure 1d*) and its sensory responses (*Figure 3— figure supplement 2*). Instead, this asymmetry in the categorical representation stems from a task-induced population-level enhancement of Go and suppression of No-Go categorical responses, consistent with our previous findings (*Bagur et al., 2018*) in these Go/No-Go paradigms. To elaborate, No-Go sounds instruct the animal to maintain the same behavioral output (lick inhibition) as in periods of silence when spontaneous activity is measured. We therefore proposed that the alignment of the No-Go population responses with the spontaneous activity reflects the identical behavioral meaning of these two epochs. Here, we found a similar mechanism (suppression of No-Go responses) which was confined to the categorical responses extracted through linear regression. This suggests an interesting mechanism in which population dynamics could multiplex behavior-independent sensory representations and task-modulated categorical encoding. Intriguingly, this is reminiscent of population dynamics found in prefrontal cortex (PFC) (*Mante et al., 2013*), where sensory information and decision formation were represented along different neural dimensions.

In line with a role of A1 in the category build-up, we found that categorical information was degraded in error trials, which is consistent with an early mapping of individual stimulus into generic categories at the level of primary sensory areas. We thus propose that A1 has an initial contribution in the feedfoward formation of behavior-dependent categories. During behavior, higher areas would access an explicit representation of the behaviorally relevant categories by reading out the asymmetrical population-level encoding of Go and No-Go sounds in A1 (*Bagur et al., 2018*). These areas would then utilize the A1 categorical representation to amplify and strengthen the ongoing category, possibly passing or gating it to motor-related regions depending on the behavioral state (*Atiani et al., 2014*).

## Possible roles for sustained activity after Go sounds

Categorical responses changed at stimulus offset to maintain a Go-specific prolonged activity during the delay period, consistent with other studies showing choice-related activity in A1 (*Bizley et al., 2013*; *Niwa et al., 2012b*; *Niwa et al., 2012a*; *Guo et al., 2019*). We do not interpret this activity as efferent copies directly sent from motor regions (*Schneider et al., 2018*), as one would expect such motor-related activity to have shorter latencies (~100–300 ms; *Bagur et al., 2018*; *Orlandi et al., 2023*). Here, sound category was decodable throughout the entire delay period (*Figure 2a*), which does not match with short-latency efference copies. We have also found that the delay activity is not closely locked in time to licks since false alarm trials, in which the animals licked during the response window, were not decoded as hit trials during the delay period (*Figure 5b*).

Other studies investigating the role of A1 in auditory Go/No-Go tasks with a delay did not analyze neuronal responses after Go sounds (*Yu et al., 2021*; *Huang et al., 2016*), which may explain why this sustained activity has not been previously reported. Interestingly, delay activity in sensory cortices is thought not to be causally involved in the behavioral response of trained animals (causal inactivations in V1 [*Goard et al., 2016*] and A1 [*Yu et al., 2021*]). We note that this sustained activity after Go sounds can be caused by at least three different factors.

First, delay activity in A1 could be the result of a feedback signal from higher-order decision-related areas signaling the chosen category and maintaining it in memory by engaging A1 in a network of parietal and frontal areas (*Goard et al., 2016*). In this framework, previous works in the somatosensory system have shown that choice-related information flows bidirectionally between primary and secondary somatosensory cortices, with choice-related information emerging in primary sensory cortex, then fed to downstream areas that further feedback enriched choice-related information to the primary field (*Kwon et al., 2016*). Consistent with this hypothesis, we have demonstrated that trial-to-trial fluctuations of categorical responses during the *sound* period correlate with the amplitude of the *delay* categorical signal (*Figure 4b*), possibly suggesting that category-related information during the stimulus and delay periods is part of this communication loop.

A second interpretation is that delay activity is the result of motor preparation that would unfold over several hundreds of milliseconds (*Musall et al., 2019*). The pattern of activity observed during

early trials is consistent with this view, with faster build-up when the animals licked earlier. However, a similar pattern was not observed on false alarm trials, contrary to what would be predicted for this interpretation.

A last possibility is that delay activity signals reward expectation (*Chubykin et al., 2013*), a type of response which increases through learning (*Musall et al., 2019*; *Poort et al., 2015*). This delay activity would then reflect post-learning residual top-down feedback, or alternatively may be the signature of an eligibility trace, that is, a persistent activity necessary for bridging the gap between the sound and the response window (*Raybuck and Lattal, 2014*). Overall, disentangling these different interpretations will require further experiments.

## Methods

### Animals

Adult female ferrets (*Mustela putorius furo*) obtained from Marshall BioResource were used for this study. The animals were 1–3 years of age, weighing 500–800 g, and were housed in pairs or trios with a normal day–night light cycle and free access to water during weekends. Ferrets were on a water-controlled protocol in which their water intake is restricted during the weekdays. Water was delivered during behavioral sessions as a reward. To maintain a stable weight, we provided ad libitum water for 1–2 hr post behavior. The animals' weights were daily monitored and maintained at 80% of pre-experiment weight.

### Behavioral task and training

Two adult female ferrets (*M. putorius furo*) were trained on an appetitive Go/No-Go delayed categorization task and one additional ferret was used for naive recording. The animals were head-fixed in a custom-made tube during training and recording sessions, and the stimuli were presented from a calibrated earphone (Sennheiser IE800, HDVA 600 amplifier). Ferrets had to classify click trains into two categories: target (Go) and non-target (No-Go) depending on the rates of click trains. Six rates were used, from 4 to 24 Hz in 4 Hz steps, and with a category boundary fixed at 14 Hz. To ensure the dissociation between categories and stimulus rates, one animal was trained with low rates as the Go sounds, while the second animal classified high rates as the Go sounds. Clicks were monopolar, rectangular pulses of 1 ms duration with amplitude set at 70 dB sound pressure level. A trial started with a pre-stimulus silence of 0.5 s followed by a click train (Go or No-Go) of 1.1 s (*Figure 1*). The animals were trained to wait for a response window that started after a delay of 1 s following stimulus offset. A hit (lick on the response window for Go stimuli) was rewarded with 0.2 ml of water. An LED attached to the water spout emphasized the delay period in which the animal had to restrain from licking. Early trials (lick during stimulus and delay period) and false alarms (lick on the response window of a No-Go sound) were punished with a timeout of 10 s. In each session, Go and No-Go stimuli were presented in a pseudo-random manner. In the absence of delay, ferrets learned to associate the categories in 1 wk, and we then slowly increased the delay between stimulus offset and response window. It took several weeks for the ferrets to be trained on the full task structure. Initially we trained with extreme categorical stimuli (4 and 24 Hz) that were easy to learn, and after reaching a consistent performance (d′ > 1), we progressively introduced other stimuli.

### Surgery

To head restrain during training and obtain stable neurophysiological recordings, we implanted the ferrets with a stainless steel headpost. The day prior to the surgery ferrets were injected with antibiotics (Baytril, 12.5 mg/kg subcutaneous) to minimize infections arising from the surgery. On the day of surgery, ferrets were deprived of water and food 90 min prior to the surgery. After sedation (medetomidine, 0.08 mg/kg, subcutaneous), anesthesia was induced with ketamine (5 mg/kg, intramuscular). Animals were kept under deep anesthesia (1–2% isoflurane) throughout the surgery and vitals (ECG, pulse, oxygenation, and rectal temperature) were continuously monitored. We also medicated the animals with atropine sulfate (0.2 mg/kg) to stabilize salvation and control arrhythmia arising from anesthesia. Using a complete sterile procedure, the animal skull was surgically exposed by an incision to the skin along the media crest down to the neck. The temporal muscles were carefully removed from the medial crest to the beginning of the zygomatic arch and the lateral wing at the lateral end of

the nuchal crest. Using a stereotaxic apparatus, the headpost was mounted on the skull using methyl methacrylate-based dental adhesive resin cement. Stainless steel screws were anchored along the areas surrounding the auditory cortex, leaving a cavity for easy access to the auditory cortex. The typical horseshoe shape of the auditory cortex was marked with nail polish. Finally, the surrounding areas were filled with poly-methyl methacrylate-based bone cement to stabilize the implant. Antibiotics (Baytril, 12.5 mg/kg, subcutaneous) and analgesic (meloxicam, 0.05 mg/kg, oral) were administered to the animal following the surgery.

We allowed a 2-week postoperative care for the animals to recover from the surgery. Antibiotics were continued for 7 d and anti-inflammatory and analgesics were administered for 4 d. Animals were habituated to a head-restrained custom-made horizontal plastic tube a few days prior to training session. Experiments were approved by the French Ministry of Agriculture (protocol authorization: 21022) and strictly comply with the European directives on the protection of animals used for scientific purposes (2010/63/EU).

## Neurophysiological recordings

In a separate surgery, we chronically implanted 32-channel metal electrodes arrays (MEA; Pt-Ir, Micro-Probes, 8 × 4, electrode of impedance 2.5 MΩ with 0.4 μm distance between the electrodes) over the prior marked auditory cortex. We custom-designed the chronic implant with MEA inserted in a drive-shuttle system having a flexible control of the array vertical movement. The base of the drive was sealed with a stretchable silicon membrane sheet to stop flowing any residues into the drive. Before the implantation, the electrodes were moved down such that the apex popped out of the silicon membrane. Under surgical anesthesia (isoflurane 1 %), we removed the cement above the location marked during the surgery. We then performed a 4 mm × 4 mm craniotomy. This craniotomy allowed us to identify the core regions of the auditory cortex (middle ectosylvian gyrus) by visual inspection of the tip of the ectosylvian gyrus. We carefully removed the transparent dura to ensure that the array penetrated the brain without additional strain. The drive-shuttle system was placed on the brain surface using a stereotaxic apparatus and the entire system was fixed to the skull using bone cement. To minimize vibrations that could be caused by shocks on the drive, the chronic implant was enclosed in a custom-made tube cemented to the implant. Immediately after the surgery we slowly lowered the electrodes and observed physiological activity, allowing us to verify the electrodes moved inside the brain.

Each recording session consisted of passive and active sessions. Recordings were performed head-fixed in a soundproof chamber. In the passive sessions, the water spout was removed. Continuous electrophysiological recordings were digitized (31,250 Hz), amplified (15,000×), and band-passed between 300 Hz and 7000 Hz using a digital acquisition system (Blackrock Cereplex). Band-passed signals were monitored online and units (including multi- and single-) were identified by spikes crossing a threshold of 3 SD of baseline noise. The data acquisition was done using an open-source suite MANTA v. 1.0 (*Englitz et al., 2013*). We used a custom-made open-source software Behavioral Auditory PHYsiology (BAPHY) written in MATLAB for sound delivery, recording, behavioral monitor and online analysis.

To identify units, we presented band-pass noise (0.2 s duration, 1 octave bandwidth) and pure tone stimuli to the animal using earphones (Sennheiser IE 800). Primary auditory cortical responses were identified by analyzing tuning properties to 100 ms tone pips of random frequencies spanning 4 octaves and temporally orthogonal ripple combinations (STRF) (*Depireux et al., 2001*). A1 responses show sharp tuning to random tones and single-peak, short-latency STRFs (*Atiani et al., 2014*; *Fritz et al., 2003*; *Elgueda et al., 2019*). Finally assessing the STRFs, we continued with the experimental protocols to record the neuronal responses.

**Table 1.** Number of recorded units.

| Ferret | Recorded Units | Units used for population analysis | Sessions |
|---|---|---|---|
| Ferret P | 575 | 395 | 35 |
| Ferret T | 241 | 203 | 39 |

## Unit identification and spike sorting

We performed offline spike sorting on thresholded signals using PCA-based customized spike sorting routines written in MATLAB. Single- and multi-unit responses were identified by spiking shape and manually adjusting the PCA clusters (*Fritz et al., 2003*). A total of 575 (ferret P) and 241 (ferret T) multi-units were identified and used for further analysis (*Table 1*). Spike sorting was done on concatenated passive and active sessions. We obtained 11.3 ± 4.9 neurons per session (± SD; n = 35 sessions) for ferret P and 5.2 ± 3.2 neurons per session (± SD; n = 39 sessions) for ferret T.

## Data analysis

### Preprocessing

Offline data analysis was performed using custom-written scripts in MATLAB (R2016a). All units were preprocessed to identify stable units that were kept under recording for both the active and passive sessions. We used a firing rate-based threshold to find stable units with non-zero firing rate for >80% of the trials, and the difference between time-averaged maximum and minimum firing rates is less than tenfold across trials. We only analyzed units with >2 spikes/s firing rate. This procedure yielded 395 units in ferret P and 203 units in ferret T. Spike counts were constructed for 100 ms non-overlapping time bins and thus used for further analysis. All population analyses were done at the single-session level, and therefore individual sessions were used as samples in statistical tests. This also allowed us to use all trials in each session, despite the difference in the number of correct trials across sessions due to variable behavior.

### Neurons with sustained activity during the delay period

To identify neurons with sustained activity during the delay, we z-scored spike counts from the baseline period (500 ms pre-stimulus) and then applied a threshold of 1.5 to the z-scored firing rate averaged over the delay period.

### Modulation index

For each unit, the modulation index of activity for quantifying modulation of delay activity was computed using

$$MI = \frac{Delay - Spont}{Delay + Spont}$$

where *Delay* is the firing rate during the delay and *Spont* is the spontaneous activity over the pre-stimulus period.

### Population decoding

In each recording session, we constructed time-based binary linear discriminant classifiers (*Bagur et al., 2018*; *Bishop, 2006*; *Meyers et al., 2008*) to decode stimulus categories (Go vs No-Go) with 100 ms binning. In brief, for each recording sessions, population vectors were constructed at each time bin and trained with equal number of random Go and No-Go trials ($V_t^{Go}, V_t^{No-Go}$). Then the population decoding vector is given by

$$W_t = V_t^{Go} - V_t^{No-Go}$$

where $V_t^{Go}, V_t^{No-Go}$ are trial-averaged population vectors for Go and No-Go categories, respectively. The threshold is defined by

$$b_t = \frac{-\left(W_t x V_t^{Go} + W_t x V_t^{No-Go}\right)}{2}$$

For each test trial population vector $V_t^{test}$, the linear discriminant function is given by

$$Y_t = W_t \times V_t^{test} + b_t$$

The test trial $V_t^{test}$ is assigned to the Go category if $Y_t \geq 0$ and to the No-Go category otherwise. The assigned category is compared to the ground truth and the proportion correct defines the accuracy of the classifier. Cross-validation was performed 200 times by randomly choosing train (70% of data) and test trials (30% of data).

Random classifier performance was obtained with 200 label shuffling permutations (the lower bound for the p-value being 1/200 = 0.005) and the comparison was statistically evaluated through permutation tests, comparing the actual performance with the chance-level distribution. Unless mentioned otherwise, all analyses were done using only correct trials (correct rejection No-Go trials and hit Go trials). Temporal evolution of the decoder was computed as the correlation between decoding weight vectors at one time bin against others, and time points below-chance correlation values are shown in gray in *Figure 2b*. We also projected the trial-averaged activity of test trials onto the unit decoders trained at sound and delay period as shown in *Figure 2—figure supplement 4*.

## Linear regression

A linear decoder trained on classifying categories inherently mixed sensory and category information to decode categories. To disentangle the contributions from sensory features and categories in the population code, we opted for linear regression models to capture the unique contribution of each feature.

At each time bin, the design matrix for linear regression consists of sensory (click rates 4, 8, 12, 16, 20, and 24 Hz) and category (–1 for No-Go and +1 Go) regressors as follows:

$$r_i\left(t\right) = \beta_0\left(t\right) + \beta_1\left(t\right) * Sensory + \beta_2\left(t\right) * Category$$

where $\beta_{1,2}\left(t\right)$ are the regressor weights, $\beta_0\left(t\right)$ a constant, and *Sensory* and *Category* are the regressor values.

The model was fitted for each neuron on correct trials (hit Go trials and correct rejections No-Go trials). We used ridge regression to minimize overfitting and diminish the impact of correlated variables. The ridge parameter is calculated using a cross-validated marginal maximum likelihood method (*Karabatsos, 2018*). To fit the model, we used the ridgeMML function from http://churchlandlab.labsites.cshl.edu (*Musall et al., 2019*).

In order to make sure that the sensory and category regressor weights were not contaminated by other task variables, we included lick and reward variables in a separate model fitted on correct and incorrect trials. At each time bin, the lick regressor consisted of either 1 or 0 corresponding to the presence or absence of licks. During the hit trials, the reward regressor was set to 1 the time bins in response window when the animal licked and was 0 otherwise. We found that the category and sensory axes were similar in the two-regressor and the four-regressor models (*Figure 2—figure supplement 5*).

## Coefficient of partial determination

To capture the contribution of each feature into neuronal activity, we computed a CPD as the fraction of variance lost by shuffling one of the features with respect to the full original model (*Fisher et al., 2019*). Doing so, CPD captured the unique contribution of that feature (*Musall et al., 2019*). We thus fitted reduced linear models with one of the variables being shuffled in the design matrix in order to destroy the contribution arising from that particular task variable. We used fivefold cross-validation to compute the mean square error (MSE) for the full and reduced models. CPD was defined as

$$CPD = \frac{MSE_{red} - MSE_{full}}{MSE_{red}}$$

where $MSE_{red}$ is obtained by cross-validating the following reduced models with fivefold cross-validation:

- For estimating the unique contribution of sensory regressor:

$$r_i\left(t\right) = \beta_0\left(t\right) + \beta_1\left(t\right) * Sensory_{shuffled} + \beta_2\left(t\right) * Category$$

- For estimating the unique contribution of category regressor:

$$r_i\left(t\right) = \beta_0\left(t\right) + \beta_1\left(t\right) * Sensory + \beta_2\left(t\right) * Category_{shuffled}$$

Time windows for computing average CPD in each period were 0–1.1 s for the stimulus period, 1.1–2.1 s for the delay period, and 2.1–3.1 s for the response window period.

## Projections of population activity onto sensory and category neural axes

We assessed the time evolution of population activity along sensory- and category-related neural axes defined by the linear regression (*Figure 3*). This was done by projecting baseline-corrected population activity specific to the feature of interest onto category or sensory regressor weights. Therefore, any deviation from zero represents the deviation of population activity along the regression axis away from the projection of baseline activity.

Regression weights for the active sensory and category neural axes were correlated during the stimulus period ($\rho$ = 0.37, p<10$^{-2}$, n = 35 sessions in ferret P; $\rho$ = 0.17, p=0.04, n = 38 sessions in ferret T; Pearson correlation). We made sure to separate the contribution of each variable before projecting on the regression weights. For restricting the projections of activity along each axis (say feature X, which could be category), we first subtracted the response predicted by the model for the other regressor (say feature Y, in this case sensory: $\beta_1\left(t\right) * Sensory$) from the single-trial population activity. We then projected the residual population activity onto the regression weights of feature X. Projections are therefore computed as follows:

$$Proj_{1/2}\left(t\right) = \frac{\beta_{1/2}\left(t\right)}{\left\|\beta_{1/2}\left(t\right)\right\|} * \left(r_{test}\left(t\right) - \beta_{2/1}\left(t\right) * Category\right)$$

where 1/2 corresponds to weights for sensory and category regressors.

Before projecting, the test trial activity at each time bin was baseline-corrected, so that projections of baseline activity lie at zero. We could then project population activity corresponding to individual click rates (*Figure 3a–d*) or averaging across click trains of the same category (*Figures 3e and 4*). Time windows for computing average projection axis in each period were 0.2–0.9 s for the stimulus period and 1.3–2.1 s for the delay period.

## Categorization and sensory indices

We quantified the amount of category information present in the projections using a category index (CI). CI is an index comparing the distance between categorical responses of stimulus pairs within and between categories:

$$CI = d\left(between\right) - d\left(within\right)$$

$d\left(within\right)$ was averaged over all possible successive pairs belonging to the same category (4–8 Hz, 8–12 Hz, 16–20 Hz, and 20–24 Hz) while $d\left(between\right)$ was measured at the category boundary (12–16 Hz).

Similarly, a sensory index (SI) measured the distance between the projections of successive click rates within the same category (4–8 Hz, 8–12 Hz, 16–20 Hz, and 20–24 Hz) onto the sensory regressor weights:

$$SI = average\left(d\left(within\right)\right)$$

where $d\left(within\right)$ is the projection distance between successive click rates within the category.

### Statistics and data availability

We performed both population decoding and linear regression analysis session-wise with simultaneously recorded neurons. All statistics across passive and active states were done with paired *t*-tests. Correlations were linearly assessed (Pearson correlation). Unless specified otherwise, error bars showed ± 1 SEM over sessions. The codes for reproducing the analysis are available on the following repository: https://github.com/rupeshjnu/A1-Category, (copy archived at *Rupesh, 2023*). The electrophysiological data is publicly available at https://zenodo.org/records/8371084.

## Acknowledgements

We thank Joao Barbosa and Célian Bimbard for deep reading and fruitful comments. This work was supported by ANR-17-EURE-0017 and ANR-10-IDEX-0001-02, ERC 787836-NEUME and ANR-19-CE37-0016 to SAS, and ANR-JCJC-DynaMiC to YB.

## Additional information

### Competing interests
Srdjan Ostojic: Reviewing editor, *eLife*. The other authors declare that no competing interests exist.

### Funding

| Funder | Grant reference number | Author |
| --- | --- | --- |
| Agence Nationale de la Recherche | ANR-17-EURE-0017 | Rupesh K Chillale Shihab Shamma Srdjan Ostojic Yves Boubenec |
| Agence Nationale de la Recherche | ANR-10-IDEX-0001-02 | Rupesh K Chillale Shihab Shamma Srdjan Ostojic Yves Boubenec |
| Agence Nationale de la Recherche | ANR-JCJC-DynaMiC | Yves Boubenec |
| H2020 European Research Council | ERC 787836-NEUME | Shihab Shamma |
| Agence Nationale de la Recherche | ANR-19-CE37-0016 | Shihab Shamma |

The funders had no role in study design, data collection and interpretation, or the decision to submit the work for publication.

### Author contributions
Rupesh K Chillale, Conceptualization, Resources, Data curation, Formal analysis, Validation, Investigation, Visualization, Methodology; Shihab Shamma, Validation, Investigation, Visualization, Writing - review and editing; Srdjan Ostojic, Conceptualization, Supervision, Funding acquisition, Validation, Investigation, Methodology, Project administration, Writing - review and editing; Yves Boubenec, Conceptualization, Resources, Formal analysis, Supervision, Funding acquisition, Validation, Investigation, Visualization, Methodology, Writing - original draft, Project administration, Writing - review and editing

### Author ORCIDs
Rupesh K Chillale http://orcid.org/0000-0002-0858-9660
Srdjan Ostojic http://orcid.org/0000-0002-7473-1223
Yves Boubenec http://orcid.org/0000-0002-0106-6947

### Ethics
Experiments were approved by the French Ministry of Agriculture (protocol authorization: 21022) and strictly comply with the European directives on the protection of animals used for scientific purposes (2010/63/EU).

### Decision letter and Author response
Decision letter https://doi.org/10.7554/eLife.85706.sa1
Author response https://doi.org/10.7554/eLife.85706.sa2

## Additional files

### Supplementary files
• MDAR checklist

## Data availability

Data is available on Zenodo, https://doi.org/10.5281/zenodo.8371083.

The following dataset was generated:

| Author(s) | Year | Dataset title | Dataset URL | Database and Identifier |
|---|---|---|---|---|
| Chillale RK, Boubenec Y | 2022 | Dynamics and maintenance of categorical responses in primary auditory cortex during task engagement | https://doi.org/10.5281/zenodo.8371083 | Zenodo, 10.5281/zenodo.8371083 |

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
