## [Editor Report]

This study provides an important contribution to our understanding of the neural basis for the categorical perception of sounds. Although the number of animals included is small, solid evidence is presented to show how categorical information emerges in the ferret primary auditory cortex following sound presentation and persists until a behavioral response is made. The work will be of interest to neuroscientists interested in the neural representation of task–related variables in sensory cortex during decision–making tasks.

---

## [Decision Letter]

**Decision letter after peer review:**

Thank you for submitting your article "Dynamics and maintenance of categorical responses in primary auditory cortex during task engagement" for consideration by *eLife*. Your article has been reviewed by 3 peer reviewers, and the evaluation has been overseen by a Reviewing Editor and Andrew King as the Senior Editor. The reviewers have opted to remain anonymous.

Essential revisions:

1. More information on how the linear regression analysis used to separate auditory and category–related activity was carried out is needed. This analysis should also be expanded by including other task–relevant parameters (licking, reward, uninstructed movement) in order to provide stronger evidence that the changes in A1 activity represent a categorical response rather than a premotor response or signals related to reward expectation. Related to this point, the reviewers were concerned that the effects reported are very small.

2. More details of the population decoding are needed and there are many places where important methodological and other information is missing (see individual reviewer comments).

3. The naïve animal included in the study was not thought to be an ideal control. Evidence that A1 neurons encode learned categories would be stronger if the comparison could be made with ferrets that have been pre–trained on the task structure. i.e. are performing the task, licking, receiving rewards, etc, but have not learned the stimulus categories. If possible, please provide this more appropriate control.

*Reviewer #1 (Recommendations for the authors):*

In this study, Chillale et al. investigate auditory cortex population dynamics during sound categorization. The authors train ferrets in a delay go/nogo task and record neuronal activity with multielectrode arrays. The main finding is that learned categories are encoded in A1 population activity, both in a task–engaged and a passive setting. The task–engaged category representation persists during the delay, but its activity pattern is uncorrelated with that during stimulus presentation. Further analyses show that the 'nogo' category sound representation is suppressed during task engagement and that sensory encoding during stimulus presentation is degraded when the ferret makes a mistake. The authors conclude that the A1 category representation is an early contributor to auditory categorization.

Overall, the study uses an elegant task design and provides important insights into the encoding of learned categories early in sensory processing. The task, having a delay period between stimulus presentation and response window, allows the authors to address how the neuronal representation evolves in absence of the stimulus and before the onset of motor behavior. The finding that neural activity during stimulus presentation and delay periods encodes learned categories, but is uncorrelated, is particularly intriguing, as it suggests complex circuit dynamics beyond sensory processing in A1. In addition, the use of stimuli varying in click rates rather than, the more common, sound frequency makes the results less sensitive to sampling biases (i.e. through tonotopy). In general, the focus of this study on characterizing population dynamics rather than describing single–neuron responses provides an interesting contrast to previous studies on sound categorization.

I have two suggestions for how the conclusions could be strengthened and presented in a better interpretable way. The first suggestion concerns the approach to disentangle stimulus from the category–related activity and its description in the manuscript, and the second is regarding the question of whether the category representation is indeed a learned sensory representation, or could reflect other task–related aspects. I will detail both below.

1. The authors use a linear regression model to disambiguate stimulus–specific and category–specific population activity. They then use the learned regressor weights to project population activity onto a category coding dimension. However, from the manuscript, it cannot be inferred whether this model can isolate sensory– and category–related activity. The methods state that the sensory regressor assumes a linear relationship between neural activation and click rate. I am not convinced that this is always the case in the auditory cortex. In addition, since there is a stable mapping between click rate and category identity, the two regressors are not independent, which can lead to unreliable estimation of regression weights and lower statistical power (multicollinearity). This problem potentially extends to the method for calculating the projection of activity using the model weights, the projection axes could be not independent or orthogonal. Therefore, it is hard to determine if the category encoding projection is unique to category encoding. The model also lacks other behavioral parameters, like licking, reward, and uninstructed movement parameters, that therefore go unaccounted for.

In order to address these concerns, the authors should amend the regression model with the other task–relevant parameters, and describe in more detail how it was constructed and fit (e.g. showing a design matrix and/or schematic or toy model illustrating how projections were made), report on its performance (e.g. Rsquared) and show that their choice of regression parameters, and method of calculating projections, leads to separable dimensions/projection axes. In addition, the authors can explore whether single neuron click rate tuning (in naive animals) indeed does not give rise to any 'categorical response'.

2. The authors compare their findings in trained animals to passive recordings in one naive animal. This comparison is used to argue for a learned component of the category representation. However, this finding is (to some degree) confounded with other learned task–related behaviors/associations that covary with the learned categories.

The conclusions of this study would be vastly strengthened if a comparison was done with "naive" animals that have been pre–trained on the task structure. i.e. are performing the task, licking, receiving rewards, etc, but have not learned the stimulus categories (yet). I would find it very useful to know whether the 'categorical response' is absent in behaving animals that did not learn the category association. This approach would also allow the authors to more directly address the specificity of the 'nogo' suppression with learned behavioral inhibition and to compare the influence of motor planning on the category representation during the delay period.

In addition to the two main suggestions, I have some more detailed comments on how the authors could improve their manuscript.

1. Clarity and precision of method descriptions. Several key methods are hard to follow, I listed some questions in the bullet points below:

As mentioned above, the model description is unclear, what is t, time, or trial?

When describing trial–averaged data, what was averaged, frames in a trial, or trial repetitions?

What linear decoder was used?

Was the whole task done head–fixed?

The section on surgical procedures is imprecise, were multiple craniotomies performed? It says the craniotomies were made into cement. How did the craniotomies allow for the visual identification of A1 regions?

2. Clarity and precision of figure legends and statistics descriptions. Often it is not clear from text or legends if a plot shows data from one or both animals (e.g. Figure 2b, but also others), what information the error bars display (e.g. Figure 1b,c Figure 2c), to which time–bin decoders are trained/regressors are fit (e.g. Figure 4a), and what certain figure elements, like black bars, represent (e.g. Figure 3f, 4d). Some statistical tests are only reported in text and not in the respective figure and legends, some vice versa (eg. Figure 2e). Some labels are switched (Figure 2b).

3. In the analysis of error trials, it seems both misses and false alarms are combined. However if the hypothesis is that in nogo trials the category representation, and hence behavioral action, is suppressed, shouldn't there be an interesting difference between misses and false alarms? This should be explored to strengthen the interpretation of the results.

4. The discussion and clarity of the interpretations of results could be improved, specifically regarding the following questions:

Is the interpretation of the uncorrelated stim and delay period decoders (Figure 2b) that different neurons are responsible for the encoding? Are there alternative interpretations?

Can overall changes in population activity (more firing, attention,..) explain the trial–to–trial correlation of stimulus period and delay period activity (Figure 4b)?

It would be very interesting to discuss in more detail how this study's findings relate to the observations on a single neuron level (Xin et al., 2019).

The direction of argumentation is unclear when it comes to the question of whether the feedback–related activity can explain the results. The authors could also consider work in the somatosensory system (e.g. Yang et al., Nat Neuro, 2016).

If underlying rotational dynamics are suspected, would those not lead to a gradual shift in correlation rather than an abrupt switch?

5. The authors state in the first paragraph of the discussion that encoding of behavioral categories was not observed in the naive animal. Does this refer to Figure S4? I do not see any data supporting this claim.

*Reviewer #2 (Recommendations for the authors):*

The authors trained two ferrets to discriminate high and low–rate click trains with target and non–target categories each comprised of three distinct rates with one animal trained to treat fast rates as targets and the second to treat slow rates as targets. In each case animals are rewarded for licking targets and required to refrain from licking non–target trials. Additional complexity is imposed by ferrets being required to withhold their response until a delay period (indicated with an LED) ends. Requiring that animals wait for this extra delay period offers the potential to parse out sensory, categorical, and motor aspects of the neural response.

Neural recordings are made in passive and active configurations for every neuron. The authors use population decoding to explore the emergence of categorical responses through the trial duration with their key finding being that categorical information exists during the stimulus, the delay period, and the response period, whereas sensory information is predominantly encoded in the stimulus period. During passive listening stimulus and (some) category information exists in the stimulus, but not the later epochs.

Broadly speaking this study confirms similar findings relating to the time course over which single neuron choice probabilities emerge in A1 (which are closely related to categorical perception in animals with good task performance, as they quantify the ability to decode the class of response across multiple stimulus values) from e.g. the Sutter lab, the Bizley lab, the Jaramillo lab.

I was left a little uncertain as to how large the conceptual advance was here, and a little unsatisfied that I don't know anything about the neural responses within the population (or even how the population response is being modelled, see comments below). For example, is the category sensitivity a consequence of a few category–tuned neurons, or is it an emergent property only visible in the population? How does the category selectivity compare to single neurons decoded over analogous time windows? What is the added value of combining information across small groups of neurons? I also don't follow the argument for why their data support that delay activity might be feedback activity (although I agree this is a perfectly plausible theoretical argument).

I have some concerns about to what extent the 'categorical' response really represents an abstraction of the stimulus class, as opposed to a premotor response or feedback from an area generating such signals. In this regard the observation that activity during error trials for all but the latest epochs of the trial is noisy / chance level rather than below chance is reassuring; however, the 'error' trials here include two categories of behaviour – both misses (where there was no motor response) and false alarms (where the motor response was early). This analysis would be more convincing if each class of response was separated out.

I would also like more information on the population decoding approaches. The authors apply a regression model which is an elegant solution to try and tease apart the confounded sensory and category information. However, the coefficient of partial determination (i.e. the variance explained by one or another factor) is tiny – on average ~ 1% of the overall variance. This begs the question of how good the linear models are in the first place, and what proportion of the explainable variance 1% accounts for (maybe it's a large fraction but without more information about model fit we can't assess this). In fact, the sensory information also shows a u–shaped function, being high during the stimulus, low in the delay then nearly as high as the stimulus period in the response window. This doesn't seem to fit with the narrative put forwards in the manuscript. There aren't sufficient details (or code) in the methods to work out what is actually modelled – what is the neural response at time t and how does it relate to the population of units (i.e. is it the average spike rate across the population or a vector of unit spike rates or a matrix of spike rates over time… ?). Without this information, it makes it very hard to understand what is being projected back onto the regression coefficients.

For the linear decoder more details (or as a minimum a reference) are needed – is it a Foffani and Moxon style Euclidean distance decoder, an SVN, or … ? I presume 2A is the result of the linear decoder? It would be nice to see something a little closer to the raw data here instead of just mean {plus minus}SD. 2B is also the linear decoder. Generally speaking, there are insufficient details in the methods for the population decoding to really understand what was run, and even less so to replicate their study. More details need to be provided here (and ideally the code released alongside the paper).

I would have liked the information about the population size to be in the Results section rather than only buried deep in the methods; the populations themselves are really quite small (mean 5 / 11 neurons in the two animals) which is useful in interpreting the modest performance of the decoder (which is clearly above chance but not that much so). Also how confident are they that all units are in A1 as the array sounds like it's quite large (potentially larger than A1) to me?

There are many places in the manuscript where it's not obvious whether the data is from one animal or both (one assumes one animal, as the figures list only a single contingency for high/low rates). The data for both animals are very clearly laid out in the supplemental material but not always well described in the main manuscript.

*Reviewer #3 (Recommendations for the authors):*

This work investigated the activity of neurons from the primary auditory cortex (A1) of ferrets performing a click–rate categorization go/no–go task or passively listening to these sounds. The authors found that the population of recorded A1 neurons shows a different firing pattern for go vs. no–go stimuli, not only during the stimulus presentation but also during a delay period before the licking response. Prediction of the go vs. no–go categories via neural decoding analysis revealed that these categories were decodable during both the stimulus and delay periods, but the population code was different between these two periods.

The authors provide clear evidence of differences in neural activity patterns for correct trials with go vs no–go stimuli. However, it is not completely clear that these observations reflect auditory categorization as the authors suggest. Most of the data presented seems consistent with alternative interpretations such as a representation of expected reward or pre–motor signals in the auditory cortex. For example: (1) the differences in neural activity between go and no–go stimuli are not present during the passive presentation (Figure 1f) when animals are presumably not licking or expecting reward; (2) the dynamics of neural activity changes consistently with movement when comparing (invalid) early licks, (invalid) late licks and (valid) hit trials (Figure 4c); and (3) the population code that enables decoding of go vs no–go stimuli changes between the stimulus presentation period and the delay period, which suggests a change in what is being represented during these periods (which could be mostly stimulus identity in the first period and motor–preparation signals in the second period). As such, the claim that neural activity reflects the categorization of the stimuli rather than the representation of other variables does not seem fully supported.

The authors try to address some of these concerns in the discussion by suggesting that motor–related activity is expected to have a short latency (~100 ms). However, from their experiments, it seems difficult to rule out that signals related to motor preparation or reward expectation (at possibly multiple latencies) are the main drivers of the observed effects.

If we define perceptual categorization as a maximization of perceptual differences between categories and a minimization of the differences within a category, investigating the neural representation of auditory categories may require a more nuanced comparison of how well one can decode stimuli within vs. across categories from neural activity.

– The manuscript would benefit from a discussion of alternative explanations related to reward expectation.

– The differences in neural activity seem compelling, but the author may want to de–emphasize the idea that these changes are associated with a neural representation of auditory categories.

– Figure 1e: because the structure can appear from random data when sorted, a supplementary figure showing neurons sorted by the delay during passive would illustrate that effects shown in this figure are not just the result of sorting for the active condition.

– It would be useful to clarify whether the "increase" associated with Figure 1f (during the delay period) is with respect to the spontaneous or sound–evoked activity.

– Clarify what "R.W" means. I don't think that is a standard acronym.

– Figure 1f: specify what period(s) of activity the modulation index refers to.

– The authors need to clarify whether the animals are head–fixed or freely moving during training and recordings. While they mentioned "To obtain stable neurophysiological recordings we implanted the ferrets with a stainless steel headpost", it's not clear when the headpost was used since the electrodes were chronically implanted.

– Authors should also specify how sounds were delivered (Figure 1 seems to indicate the ferrets had headphones).

– Authors should be clearer about the passive stimulation sessions. Are the animals licking? are there other differences compared to the active sessions (e.g., inter–trial interval)?

– The first mention of "the naive animal" comes out of nowhere. The authors should introduce that there is a naive animal used for control experiments.

– Figure 2 caption: I don't know what "resp." means.

– Figure 4C: y–label should say "categorical".

– Figure 5a: the caption says "cyan" but it looks purple to me.

---

## [Author Response]

Essential revisions:1. More information on how the linear regression analysis used to separate auditory and category–related activity was carried out is needed. This analysis should also be expanded by including other task–relevant parameters (licking, reward, uninstructed movement) in order to provide stronger evidence that the changes in A1 activity represent a categorical response rather than a premotor response or signals related to reward expectation. Related to this point, the reviewers were concerned that the effects reported are very small.

We thank the reviewers for their detailed suggestions that have helped us improve the manuscript. Briefly, we have made several main edits to the manuscript and provided the clarifications as requested by the reviewers. For instance, we have amended our linear regression model to include more task variables (licking and reward), and have added a supplementary figure describing this analysis (new Figure 2—figure supplement 5). The procedure for linear regression using more task variables, its design matrix and the regularization used were added with more information to the methods section. The percentage of variance explained is small and the same has been observed in other studies involving similar procedures, for example, Musal et al., 2019. We provide some modeling results to discuss this latter point (see Figure “CPD for category regressor as a function of noise strength” in response to Reviewer 2).

2. More details of the population decoding are needed and there are many places where important methodological and other information is missing (see individual reviewer comments).

We added an extensive description of the population decoding and other technical points to the manuscript. We have greatly revised and improved the method sections to contain all details required by the reviewers.

3. The naïve animal included in the study was not thought to be an ideal control. Evidence that A1 neurons encode learned categories would be stronger if the comparison could be made with ferrets that have been pre–trained on the task structure. i.e. are performing the task, licking, receiving rewards, etc, but have not learned the stimulus categories. If possible, please provide this more appropriate control.

We agree with this comment. However, our study’s main focus was to disentangle the task variables, especially sensory and categorical, and to investigate their population-level coding to understand their link with categorization. We introduced a long delay period after the stimulus to study how categories emerge and are maintained in a goal-directed behavior. Studying the evolution of categorical responses in A1 through learning is definitely one of our future goals. Consequently, adding a naive animal that was pre-trained on the task structure is beyond the scope of this study.

Reviewer #1 (Recommendations for the authors):In this study, Chillale et al. investigate auditory cortex population dynamics during sound categorization. The authors train ferrets in a delay go/nogo task and record neuronal activity with multielectrode arrays. The main finding is that learned categories are encoded in A1 population activity, both in a task–engaged and a passive setting. The task–engaged category representation persists during the delay, but its activity pattern is uncorrelated with that during stimulus presentation. Further analyses show that the 'nogo' category sound representation is suppressed during task engagement and that sensory encoding during stimulus presentation is degraded when the ferret makes a mistake. The authors conclude that the A1 category representation is an early contributor to auditory categorization.Overall, the study uses an elegant task design and provides important insights into the encoding of learned categories early in sensory processing. The task, having a delay period between stimulus presentation and response window, allows the authors to address how the neuronal representation evolves in absence of the stimulus and before the onset of motor behavior. The finding that neural activity during stimulus presentation and delay periods encodes learned categories, but is uncorrelated, is particularly intriguing, as it suggests complex circuit dynamics beyond sensory processing in A1. In addition, the use of stimuli varying in click rates rather than, the more common, sound frequency makes the results less sensitive to sampling biases (i.e. through tonotopy). In general, the focus of this study on characterizing population dynamics rather than describing single–neuron responses provides an interesting contrast to previous studies on sound categorization.

We thank the reviewer for the overall positive feedback and we address the specific comments below.

I have two suggestions for how the conclusions could be strengthened and presented in a better interpretable way. The first suggestion concerns the approach to disentangle stimulus from the category–related activity and its description in the manuscript, and the second is regarding the question of whether the category representation is indeed a learned sensory representation, or could reflect other task–related aspects. I will detail both below.1. The authors use a linear regression model to disambiguate stimulus–specific and category–specific population activity. They then use the learned regressor weights to project population activity onto a category coding dimension. However, from the manuscript, it cannot be inferred whether this model can isolate sensory– and category–related activity. The methods state that the sensory regressor assumes a linear relationship between neural activation and click rate. I am not convinced that this is always the case in the auditory cortex. In addition, since there is a stable mapping between click rate and category identity, the two regressors are not independent, which can lead to unreliable estimation of regression weights and lower statistical power (multicollinearity). This problem potentially extends to the method for calculating the projection of activity using the model weights, the projection axes could be not independent or orthogonal. Therefore, it is hard to determine if the category encoding projection is unique to category encoding. The model also lacks other behavioral parameters, like licking, reward, and uninstructed movement parameters, that therefore go unaccounted for.In order to address these concerns, the authors should amend the regression model with the other task–relevant parameters, and describe in more detail how it was constructed and fit (e.g. showing a design matrix and/or schematic or toy model illustrating how projections were made), report on its performance (e.g. Rsquared) and show that their choice of regression parameters, and method of calculating projections, leads to separable dimensions/projection axes. In addition, the authors can explore whether single neuron click rate tuning (in naive animals) indeed does not give rise to any 'categorical response'.

These are all fair points. We address below the reviewer’s suggestions one by one. First, we provide more details on how the regression is designed to deal with multicollinearity. Furthermore, we assess effects of co-linearity using a toy model. We then verify that including more task variables (licking and reward) in the regression model does not significantly change the sensory and category axes. Lastly, we show in the naive animal that the choice of the sensory axis is relevant for describing a purely sensory response, and that the category regressor is, as expected, irrelevant for fitting the naive response.

Multicollinearity:

The design matrix for linear regression consists of sensory (click rates 4, 8, 12, 16, 20, 24 Hz) and category (-1 for No-Go and +1 Go) regressors. Inherently to our experimental paradigm, sensory and category regressors are correlated (ρ=0.88 n=35 sessions in Ferret P; ρ=0.84, n=38 sessions in Ferret T; Pearson correlation). We performed regularization during the regression procedure to reduce the impact of correlated variables (Karabotos et al., 2017). Furthermore, we made sure to discard the contribution of other regressors when we projected onto a regression axis.

Ridge regression**.** We used ridge regression to minimize the overfitting of the model and diminish the impact of correlated variables. The ridge parameters are calculated using the marginal maximum likelihood method from Karabotos et al., 2017. The efficacy of this method is tested using a model of a hypothetical neuron tuned to both sensory and category features. We simulated its neural response to the actual stimuli presented during a recording session and generated spiking activity as follows:

β0(t)=β1(t)∗Sensory=β1(t)∗Category=ϵ∗η(0,1), where *r_i_* (*t* )=0 for *t*<0 and *t*>1.1, *β*_1_(*t* )=0 and *β*_1_(*t* )=1 otherwise and *β*_2_(*t* )=0 for *t*<0 and *β*_2_(*t* )=1 otherwise. (0,1) is the gaussian noise and is the noise strength. For a noiseless ɳ ***ε*** neuron, = 0. By design, this simulated neural response has both sensory and category ***ε*** contributions in the time period t=0 to t=1.1 during the stimulus, and only category for t >1.1 (panels a and b Author response image 1). We then applied our linear regression model to compute the CPD. As shown in panels c and d in (Author response image 1) , the regression captured both sensory and category contributions during the stimulus period, as well as only a category contribution during the delay period. This shows that our model was able to disentangle both sensory and category contributions in the neuronal activity.

**Author response image 1. sa2fig1:** Testing the ability of the regression procedure to disentangle sensory and category contributions with modeled neurons. A model generated spike counts from both sensory and category regressors during the sound period and from the category regressor during the delay period (see text). The number of trials, number of neurons and sessions were the same as in one example session of Ferret P ( = 0.5). ***ε*** (**a**) Scatter plot of a model neuron, organized by click rates. (**b**) Average response of all neurons in the model as a function of click rate. (**c**) CPD for the sensory and category regressors. (**d**) Category regressor weights recovered from the model. (**e**) Sensory regressor weights recovered from the model.

Projection on sensory and category axes. We found that the sensory and category neural axes were correlated during the stimulus period of the task (ρ=0.37, p<10^-2^, n=35 sessions in Ferret P; ρ=0.17, p=0.04, n=38 sessions in Ferret T; Pearson correlation). We made sure to separate the contribution of each variable before projecting onto the regression weights. To restrict the projections of activity along each axis (say feature X, that could be category), we first subtracted the response predicted by the model for the other regressor (say feature Y, in this case sensory: *β*_1_(*t* )∗ *Sensory*) from the single-trial population activity. We then projected the residual population activity onto the regression weights of feature X. This is now explicitly described in the Methods:

“We assessed the time-evolution of population activity along sensory- and category-related neural axes defined by the linear regression (Figure 3). This was done by projecting baseline corrected population activity specific to the feature of interest onto category or sensory regressor weights. Therefore, any deviation from zero represents the deviation of population activity along the regression axis away from the projection of baseline activity.

Regression weights for the active sensory and category neural axes were correlated during the stimulus period (ρ=0.37, p<10^-2^, n=35 sessions in Ferret P; ρ=0.17, p=0.04, n=38 sessions in Ferret T; Pearson correlation). We made sure to separate the contribution of each variable before projecting on the regression weights. For restricting the projections of activity along each axis (say feature X, that could be category), we first subtracted the response predicted by the model for the other regressor (say feature Y, in this case sensory:) from the single-trial population activity. We then projected the residual population activity onto the regression weights of feature X. Projections are therefore computed as follows:

Proj1/2(t)=β1/2(t)||β1/2(t)||∗(rtest(t)−β2/1(t)∗Category),

Linear regression with more task variables:

To make sure that the sensory and category axes captured by the regression were not contaminated by other task variables, we supplemented the design matrix with lick and reward variables (only during the active state) and fitted this expanded regression model. At each time bin, the lick regressor consisted of either 1 or 0 corresponding to presence or absence of licks. Reward regressor also consisted of either 0 (no reward) and 1 (reward, Hit trials) in the response window. Figure 2—figure supplement 5 shows the CPD corresponding to each task variable in both animals.

It is clear that including more task variables preserves the results described in Figure 3, i.e. that A1 population activity encodes categories during the sound and delay periods upon task engagement. We then verified that the regression weights obtained with the original model of the paper, and with this updated model, were consistent. Indeed, we found high correlation values between the category weights in both models, for both animals and both the sound and delay periods (see Author response image 2).

**Author response image 2. sa2fig2:** Correlation between regressors with two and four regressor values Distributions of correlation values between category weights independantly obtained from each of the two regression models (with either two [sensory and category] or four [sensory, category, lick and decision] regressors) (n=35 sessions in Ferret P; n=38 sessions in Ferret T).

Single neuron tuning for naive and trained animals:

We have examined the single-unit tuning curves in the naive animal. The left panel shows a example neuron and the middle panel is the average tuning response of all neurons, demonstrating a linear relationship between firing rate and stimulus click rate. Overall, neural responses do not show any categorical responses. This is confirmed by the non-significant Coefficient of Partial Determination for the category regressor in the naive animal (Figure 2i).

To offer more details about the regression, we now show different example neurons exhibiting robust sensory (Figure 2c) and category tuning (Figure 2d) during the stimulus period, or category tuning during the delay period (Figure 2e). Blue solid curves are the neural responses and the magenta (sensory) and dark green (category) are the fits using only sensory and category weights while the dashed line is the linear fit using both regressors.

**Author response image 3. sa2fig3:** (**a**) Example neuron raster plot with trials sorted by click rates. (**b**) Average tuning curve over all neurons (shaded area shows 1 SEM; n=71 neurons).

This is now in the Results section:

“Single-neuron activity showed linear relationship with click rates (example in Figure 2c) or robust categorical encoding during and after the stimulus (Figure 2d,e).“

2. The authors compare their findings in trained animals to passive recordings in one naive animal. This comparison is used to argue for a learned component of the category representation. However, this finding is (to some degree) confounded with other learned task–related behaviors/associations that covary with the learned categories.The conclusions of this study would be vastly strengthened if a comparison was done with "naive" animals that have been pre–trained on the task structure. i.e. are performing the task, licking, receiving rewards, etc, but have not learned the stimulus categories (yet). I would find it very useful to know whether the 'categorical response' is absent in behaving animals that did not learn the category association. This approach would also allow the authors to more directly address the specificity of the 'nogo' suppression with learned behavioral inhibition and to compare the influence of motor planning on the category representation during the delay period.

This is a fascinating remark. The proposed paradigm would allow one to investigate how the link between stimulus and reward possibly reshapes the reward-elicited activity in auditory cortex. However, our study’s main focus was to focus on the stimulus to category transduction.

Therefore, adding a naive animal that was pre-trained on the task structure is beyond the scope of this study. Nevertheless, studying the evolution of category responses in A1 through learning is definitely a key future goal.

In addition to the two main suggestions, I have some more detailed comments on how the authors could improve their manuscript.1. Clarity and precision of method descriptions. Several key methods are hard to follow, I listed some questions in the bullet points below:As mentioned above, the model description is unclear, what is t, time, or trial?

We apologize for this. We have added a full description of the linear regression we performed and how we calculated the projections:

“A linear decoder trained on classifying categories inherently mixed sensory and category information to decode categories. To disentangle the contributions from sensory features and categories in the population code, we opted for linear regression models to capture the unique contribution of each feature.

At each time bin, the design matrix for linear regression consists of sensory (click rates 4, 8, 12, 16, 20, 24 Hz) and category (-1 for No-Go and +1 for Go) regressors, as follows:

r_i_ (t )=β_0_(t )+β_1_(t )∗Sensory+β_2_(t )∗Category

where β_1,2_ (t ) are the regressor weights, β_0_ (t) a constant, Sensory and Category are the regressor values.

The model was fitted for each neuron on correct trials (hit Go trials and correct rejections No-Go trials). We used ridge regression to minimize overfitting and diminish the impact of correlated variables. The ridge parameter is calculated using a cross-validated marginal maximum likelihood method^42^. To fit the model, we used the ridgeMML function from http://churchlandlab.labsites.cshl.edu^16^.

In order to make sure that the sensory and category regressors weights were not contaminated by other task variables, we included lick and reward variables in a separate model fitted on correct and incorrect trials. At each time bin, the lick regressor consisted of either 1 or 0 corresponding to presence or absence of licks. During the hit trials, the reward regressor was set to 1 the time bins in response window when the animal licked and was 0 otherwise. We found that the category and sensory axes were similar in the 2-regressor and the 4-regressor models (Figure 2—figure supplement 5).”

When describing trial–averaged data, what was averaged, frames in a trial, or trial repetitions?

We computed the average over *trial repetitions*. This is now clearly stated:VtGo,VtNo−Go

are trial-averaged population vectors for Go and No-Go categories respectively.”What linear decoder was used?

We used linear discriminant classifiers as prescribed in Bagur et al., 2018. Binary linear classifiers were trained to discriminate between Go and No-Go categories (Supplementary Figure 3 in Bagur et al., 2018). We now expanded and clarified the section about decoding in the methods:

“In each recording session, we constructed time-based binary linear discriminant classifiers^14,40,41^ to decode stimulus categories (Go vs No-Go) with 100 ms binning. In brief, for each recording sessions, population vectors were constructed at each time bin and trained with equal number of random Go and No-Go trials (VtGo,VtNo−Go). Then the population decoding vector is given byWt=VtGo−VtNo−Go

Where VtGo,VtNo−Go are trial-averaged population vectors for Go and No-Go categories respectively. The threshold is defined bybt=−(WtxVtGo+WtxVtNo−Go)2

For each test trial population vector V ^test^_t_ , the linear discriminant function is given by

Yt=WtxVtGo+bt

The test trial Vttest is assigned to the Go category if Y _t_ ≥0 and to the No-Go category otherwise. The assigned category is compared to the ground truth and the proportion correct defines the accuracy of the classifier. Cross-validation was performed 200 times by randomly choosing train (70% of data) and test trials (30% of data).

Random classifier performance was obtained with 200 label shuffling permutations (the lower bound for the p-value being 1/200 = 0.005) and the comparison was statistically evaluated through permutation tests, comparing the actual performance with the chance-level distribution. Unless mentioned otherwise, all analysis were done using only correct trials (correct rejection No-Go trials and hit Go trials). Temporal evolution of the decoder was computed as the correlation between decoding weight vectors at one time bin against others and time points below chance correlation values were shown as grey in Figure 2b. We also projected the trial averaged activity of test trials onto the unit decoders trained at sound and delay period as shown in Figure 2—figure supplement 4.”

Was the whole task done head–fixed?

Yes, the animals were head-fixed in both training and recording sessions. This is now stated in the Methods:

“Each recording session consisted of passive and active sessions. Recordings were performed head-fixed in a soundproof chamber. In the passive sessions the water spout was removed.”

The section on surgical procedures is imprecise, were multiple craniotomies performed? It says the craniotomies were made into cement. How did the craniotomies allow for the visual identification of A1 regions?

We have detailed the surgical procedure in the methods:

“In a separate surgery, we chronically implanted 32-channel metal electrodes arrays (MEA; PtIr, MicroProbes, 8 x 4, electrode of impedance 2.5 MΩ with 0.4 μm distance between the electrodes) over the prior marked auditory cortex. We custom-designed the chronic implant with MEA inserted in a drive-shuttle system having a flexible control of the array vertical movement. The base of the drive is sealed with a stretchable silicon membrane sheet to stop flowing any residues into the drive. Before the implantation, the electrodes are moved down such that the apex popped out of the silicon membrane. Under surgical anesthesia (isoflurane 1 %), we removed the cement above the location marked during the surgery. We then performed a 4 mm x 4 mm craniotomy. This craniotomy allowed us to identify the core regions of the auditory cortex (middle ectosylvian gyrus) by visual inspection of the tip of the ectosylvian gyrus. We carefully removed the transparent dura to ensure that the array penetrated the brain without additional strain. The drive-shuttle system was placed on the brain surface using a stereotaxic apparatus and the entire system was fixed to the skull using bone cement. To minimize vibrations that could be caused by shocks on the drive, the chronic implant was enclosed in a custom-made tube cemented to the implant. Immediately after the surgery we slowly lowered the electrodes and observed physiological activity, allowing us to verify the electrodes moved inside the brain.”

2. Clarity and precision of figure legends and statistics descriptions. Often it is not clear from text or legends if a plot shows data from one or both animals (e.g. Figure 2b, but also others), what information the error bars display (e.g. Figure 1b,c Figure 2c), to which time–bin decoders are trained/regressors are fit (e.g. Figure 4a), and what certain figure elements, like black bars, represent (e.g. Figure 3f, 4d). Some statistical tests are only reported in text and not in the respective figure and legends, some vice versa (eg. Figure 2e). Some labels are switched (Figure 2b).

Most of the results in main figures are shown for one animal such as Figure 1a-d, Figure 2, Figure 3, Figure 4a, 4c,d and Figure 5a,b. The consistency of these results in the second animal is shown in corresponding supplementary figures. We now systematically mention the number of neurons/sessions from either one or two ferrets in each figure caption. We also mention the significance bars in Figure 3f and 4d. Legends in Figure 2b have been corrected.

3. In the analysis of error trials, it seems both misses and false alarms are combined. However if the hypothesis is that in nogo trials the category representation, and hence behavioral action, is suppressed, shouldn't there be an interesting difference between misses and false alarms? This should be explored to strengthen the interpretation of the results.

The reviewer is correct in that more insight can be obtained by looking into the miss and false alarm trials separately. As shown in Figure 5—figure supplement 1, both types of error trials on the delay decoder overlap (panel B), consistent with decoding at chance level (Figure 5a). A trend of reversed trajectories on the response window decoders was observed (FA above misses in the right panel), in line with decoding below chance level in this period (Figure 5a).

4. The discussion and clarity of the interpretations of results could be improved, specifically regarding the following questions:

The discussion has been greatly expanded; we review these changes below.

Is the interpretation of the uncorrelated stim and delay period decoders (Figure 2b) that different neurons are responsible for the encoding? Are there alternative interpretations?

The stimulus and delay decoders are not correlated (Figure 2b), which is consistent with the fact that the category regression weights during stimulus and delay periods (Figure 4a) are not correlated either. We plotted the regression weights for category found during the stimulus period against the delay period. We now included this figure in the manuscript (Figure 4figure supplement 1) to unpack the correlation plot in Figure 4a.

Can overall changes in population activity (more firing, attention,..) explain the trial–to–trial correlation of stimulus period and delay period activity (Figure 4b)?

As a proxy for attention-driven changes in population activity, we compared the structure of population activity changes induced task engagement with the category neural axis extracted from the stimulus period. We found no correlation between these two axes (Author reaponse image 4), indicating that overall changes of population activity driven by attentional or arousal changes are not aligned with the category neural axis.

**Author response image 4. sa2fig4:** Distribution of correlations between the category regressor weights during the active stimulus period and the weights of state (passive/active) decoders (p =0. 125 t-test, Ferret P, n= 35 sessions).

It would be very interesting to discuss in more detail how this study's findings relate to the observations on a single neuron level (Xin et al., 2019).The direction of argumentation is unclear when it comes to the question of whether the feedback–related activity can explain the results. The authors could also consider work in the somatosensory system (e.g. Yang et al., Nat Neuro, 2016).

Thanks for suggesting these lines of discussion; we now elaborate on this point:

“Similar to a previous study in the mouse^12^, we found that the encoding of stimulus sensory properties within behavioral categories was stable across behavioral states. In Xin et al^12^, the authors found that categorical representations in A1 matched behavioral decision as early as at stimulus offset, and was maintained during several seconds. Such choice-related activity have been demonstrated to be the result of feedforward and feedback flow between primary sensory cortices and downstream regions^22,23^. Our observations are consistent with this view, as we found that perceptual choice influenced the category decoded during the response window (Figure 5a).”

If underlying rotational dynamics are suspected, would those not lead to a gradual shift in correlation rather than an abrupt switch?

Considering the changes in the discussion suggested by the reviewers, the reference to rotational dynamics did not fit anymore in this section. We therefore removed it.

5. The authors state in the first paragraph of the discussion that encoding of behavioral categories was not observed in the naive animal. Does this refer to Figure S4? I do not see any data supporting this claim.

Thanks for pointing this out. We referred to Figure 2i, in which the category CPD is for naive and is insignificant compared to that of the trained animals during the stimulus period. It is now corrected:

“First, we isolated an encoding of behavioral categories in the population-average during stimulus presentation in both task-engaged and passive listening. This was not observed in a naive animal (Figure 2i).”

Reviewer #2 (Recommendations for the authors):The authors trained two ferrets to discriminate high and low–rate click trains with target and non–target categories each comprised of three distinct rates with one animal trained to treat fast rates as targets and the second to treat slow rates as targets. In each case animals are rewarded for licking targets and required to refrain from licking non–target trials. Additional complexity is imposed by ferrets being required to withhold their response until a delay period (indicated with an LED) ends. Requiring that animals wait for this extra delay period offers the potential to parse out sensory, categorical, and motor aspects of the neural response.Neural recordings are made in passive and active configurations for every neuron. The authors use population decoding to explore the emergence of categorical responses through the trial duration with their key finding being that categorical information exists during the stimulus, the delay period, and the response period, whereas sensory information is predominantly encoded in the stimulus period. During passive listening stimulus and (some) category information exists in the stimulus, but not the later epochs.Broadly speaking this study confirms similar findings relating to the time course over which single neuron choice probabilities emerge in A1 (which are closely related to categorical perception in animals with good task performance, as they quantify the ability to decode the class of response across multiple stimulus values) from e.g. the Sutter lab, the Bizley lab, the Jaramillo lab.

The reviewer is right that our findings are related to the work of these labs. We took care of adding more references to their studies.

I was left a little uncertain as to how large the conceptual advance was here, and a little unsatisfied that I don't know anything about the neural responses within the population (or even how the population response is being modelled, see comments below). For example, is the category sensitivity a consequence of a few category–tuned neurons, or is it an emergent property only visible in the population? How does the category selectivity compare to single neurons decoded over analogous time windows? What is the added value of combining information across small groups of neurons? I also don't follow the argument for why their data support that delay activity might be feedback activity (although I agree this is a perfectly plausible theoretical argument).

We list here the conceptual advances we think we provide with respect to previous studies:

categorical activity emerged upon stimulus presentation in *trained* animals in the population average, *both* in the passive and active statesat the offset of Go sounds, sustained activity built up until lick time (over more than a second), and the encoding format of the categorical information was different between the stimulus and the delay periodsat the population-level, the representation of the No-Go sounds during the stimulus epoch became suppressed upon task engagementerror trials showed no reliable encoding of stimulus sensory features, suggesting that error came from a deficient encoding of the stimulus

We believe that the revisions made on the manuscript after the reviewers’ suggestions highlight these points much better. In particular, we clarified in the abstract, in the introduction and in the first paragraph of the discussion that the signature of category is found in the population evarge (CPD measure in Figure 2), whereas suppression of the No-Go sounds was found at the population level (when projecting on regressor weights; Figure 3).

Tuning to behavioral categories. The emergence of category-related activity during stimulus presentation was a property found in the population average. Neurons with significant CPD showed a wide range of values (see Author response image 5), indicating that this tuning was not driven by a few outliers.

**Author response image 5. sa2fig5:** 

The category selectivity was found in the population average (Figure 1—figure supplement 1c,f), and we show in Figure 2c-e the tuning curve of several example neurons, alongside with the fit to the linear regression. These neurons exhibit robust sensory (panel B) and category tuning (panel D) during the stimulus period, or category tuning during the delay period (panel E). Blue solid curves are the neural responses and the magenta (sensory) and dark green (category) are the fits using only sensory and category weights while the dashed line is the linear fit using both regressors. We also included example neurons in the manuscript (Figure 2) to exemplify the results.We now compare the category selectivity of single neurons in the stimulus and delay periods in Figure 4—figure supplement 1. Category regression weights of these two periods are overall uncorrelated, in accordance with the correlation plot in Figure 4a. We note that the weigths do not form distinct clusters.

Suppression of the population-level categorical responses. The suppression of the No-Go sound along the category axis is a collective property found at the population-level (Figure 3). Upon task engagement, the No-Go responses are more aligned with the baseline activity, leading to asymmetrical representation of the both Go and No-Go stimuli along the baseline active. We performed population-level analysis (linear decoding and projection on regression axes) at the sessions level (i.e., sessions were treated as samples) for two main reasons. First, we believe that finding effects that are consistent across *independent* sessions is an important observation that consolidates our results. Second, it allowed us to characterize how trial-to-trial fluctuations in the categorical signals correlated between the stimulus and delay epochs (Figure 4), which would not be possible with a of the neurons from all sessions.

As for the interpretations of the delay activity, we are now presenting different alternatives, including a feedback origin, but also reward anticipation.

“We note that this sustained activity after Go sounds can be caused by at least three different factors.

First, delay activity in A1 could be the result of a feedback signal from higher-order decision related areas signaling the chosen category and maintaining it in memory by engaging A1 in a network of parietal and frontal areas^33^. In this framework, previous works in the somatosensory system have shown that choice-related information flows bidirectionally between primary and secondary somatosensory cortices, with choice-related information emerging in primary sensory cortex, then fed to downstream areas that further feedback enriched choice-related information to the primary field^34^. Consistent with this hypothesis, we have demonstrated that trial-to-trial fluctuations of categorical responses during the sound period correlate with the amplitude of the delay categorical signal (Figure 4b), possibly suggesting that category-related information during the stimulus and delay periods are part of this communication loop.

A second interpretation is that delay activity is the result of motor preparation that would unfold over several hundreds of milliseconds^16^. The pattern of activity observed during early trials is consistent with this view, with faster build-up when the animals licked earlier. However, a similar pattern was not observed on false alarm trials, contrary to what would be predicted for this interpretation.

A last possibility is that delay activity signals reward expectation^35^, a type of response which increases through learning^16,36^. This delay activity would then reflect post-learning residual topdown feedback, or alternatively may be the signature of an eligibility trace, i.e. a persistent activity necessary for bridging the gap between the sound and the response window^37^. Overall, disentangling these different interpretations will require further experiments.”

I have some concerns about to what extent the 'categorical' response really represents an abstraction of the stimulus class, as opposed to a premotor response or feedback from an area generating such signals. In this regard the observation that activity during error trials for all but the latest epochs of the trial is noisy / chance level rather than below chance is reassuring; however, the 'error' trials here include two categories of behaviour – both misses (where there was no motor response) and false alarms (where the motor response was early). This analysis would be more convincing if each class of response was separated out.

The reviewer is correct in that more insight can be obtained by looking into the miss and false alarm trials separately. As shown in Figure 5—figure supplement 1, both types of error trials on the delay decoder overlap (panel B), consistent with decoding at chance level (Figure 5a). A trend of reversed trajectories on the response window decoders was observed (FA above misses in the right panel), in line with decoding below chance level in this period (Figure 5a).

This figure is now referenced in the results:

“…we found a decrease in decoding accuracy during the sound period, and a decoding performance at chance level during the delay (Figure 5a,b and Figure 5—figure supplement 1 for separate projections of false alarm and miss trials; decoding accuracy during delay for incorrect trials: p=0.99, permutation test).”

I would also like more information on the population decoding approaches. The authors apply a regression model which is an elegant solution to try and tease apart the confounded sensory and category information. However, the coefficient of partial determination (i.e. the variance explained by one or another factor) is tiny – on average ~ 1% of the overall variance. This begs the question of how good the linear models are in the first place, and what proportion of the explainable variance 1% accounts for (maybe it's a large fraction but without more information about model fit we can't assess this). In fact, the sensory information also shows a u–shaped function, being high during the stimulus, low in the delay then nearly as high as the stimulus period in the response window. This doesn't seem to fit with the narrative put forwards in the manuscript. There aren't sufficient details (or code) in the methods to work out what is actually modelled – what is the neural response at time t and how does it relate to the population of units (i.e. is it the average spike rate across the population or a vector of unit spike rates or a matrix of spike rates over time… ?). Without this information, it makes it very hard to understand what is being projected back onto the regression coefficients.

Our linear regression model was inspired by the methodology described in Musall et al., 2019. Even in the simple tasks described in this cited publicaiton, the task variables contributed less than 10% of the variance for single variable models that by definition overestimates the explained variance (for instance, Figures 4c and 7c of Musall et al., 2019). We assessed the theoretically-expected CPD value as a function of noise in the data using a hypothetical neuron tuned to both sensory and category features. We simulated its neural response to the actual stimuli presented during a recording session. We generated spiking activity as:

r_i_
*(*t *)=*β*_0_(*t *)+*β*_1_(*t *)*∗Sensory*+*β*_1_(*t *)*∗Category*+**ε***∗ɳ *( 0,1)*,

where *r_i_* (*t* )=0 for *t*<0 and *t*>1.1, *β*_1_(*t* )=0 and *β*_1_(*t* )=1 otherwise and *β*_2_(*t* )=0 for *t*<0 and *β*_2_(*t* )=1 otherwise. (0,1) is the gaussian noise scaled by the noise strength. For a noiseless neuron, ***ε*** = 0. By design, this simulated neural response has both sensory and category ***ε*** contributions during the time-period t=0 to t=1.1 during the stimulus period, and only category during t >1.1 (panels a and b Author response image 6). Then we applied our linear regression model to compute the CPD (see top sections of the response to reviewers for more details) as function of ***ε*** (see Author response image 6).

We took an example session and used the toy model to generate spike counts from the same number of neurons and trials as in the actual session. We then fitted the regression model on data generated from different noise levels ***ε***. For the actual data, we made the assumption that the noise strength can be assessed as the standard deviation σ of stimulus-evoked response. We found σ_ferret P_ = 0.94 and σ_ferret P_ = 1.41, which corresponds to CPD ~ 0.1 for the simulated neuron, consistent with the values found in the data.

**Author response image 6. sa2fig6:** Coefficient of partial determination (CPD) for category regressor as a function of noise strength.

The sensory information during the response window is difficult to sort out since licking activity is taking place during this time period. The hypothesis put forward in the manuscript mostly concerns the stimulus and delay periods, so we did not emphasize this result. Nevertheless, we expanded our linear regression model to test for an influence of licks onto population activity. We supplemented the design matrix with lick and reward variables (only during the active state) and fitted this expanded regression model. At each time bin, the lick regressor consisted of either 1 or 0 corresponding to presence or absence of licks. Reward regressor also consisted of either 0 (no reward) and 1 (reward, Hit trials) in the response window. The figure below shows the CPD corresponding to each task variable in both animals. We found that the sensory regressor explained no more neural activity during the response window, suggesting that lickrelated activity is captured by the sensory regressor.

We now expanded and clarified the section about regression in the methods:

“A linear decoder trained on classifying categories inherently mixed sensory and category information to decode categories. To disentangle the contributions from sensory features and categories in the population code, we opted for linear regression models to capture the unique contribution of each feature.

At each time bin, the design matrix for linear regression consists of sensory (click rates 4, 8, 12, 16, 20, 24 Hz) and category (-1 for No-Go and +1 Go) regressors, as follows:

r_i_ (t )=β_0_(t )+β_1_(t )∗Sensory+β_2_(t )∗Category

where *β_1,2_ (t )* are the regressor weights, β_0_ (t) a constant, Sensory and Category are the regressor values.

The model was fitted for each neuron on correct trials (hit Go trials and correct rejections No-Go trials). We used ridge regression to minimize overfitting and diminish the impact of correlated variables. The ridge parameter is calculated using a cross-validated marginal maximum likelihood method^42^. To fit the model, we used the ridgeMML function from http://churchlandlab.labsites.cshl.edu 16 .

In order to make sure that the sensory and category regressors weights were not contaminated by other task variables, we included lick and reward variables in a separate model fitted on correct and incorrect trials. At each time bin, the lick regressor consisted of either 1 or 0 corresponding to presence or absence of licks. During the hit trials, the reward regressor was set to 1 the time bins in response window when the animal licked and was 0 otherwise. We found that the category and sensory axes were similar in the 2-regressor and the 4-regressor models (Figure 2—figure supplement 5).”

For the linear decoder more details (or as a minimum a reference) are needed – is it a Foffani and Moxon style Euclidean distance decoder, an SVN, or … ? I presume 2A is the result of the linear decoder? It would be nice to see something a little closer to the raw data here instead of just mean {plus minus}SD. 2B is also the linear decoder. Generally speaking, there are insufficient details in the methods for the population decoding to really understand what was run, and even less so to replicate their study. More details need to be provided here (and ideally the code released alongside the paper).

We used binary linear classifiers, as prescribed in Bagur et al., 2018. A discriminant axis w and a decision threshold B are defined from a training set, and projections of the test set on the w axis are compared to the threshold B for predicting their labels (see Author response image 7).

**Author response image 7. sa2fig7:** 

We now expanded the population decoding in the methods section to describe more details on the classifiers are evaluation, accuracy computation and their temporal evolution:“In each recording session, we constructed time-based binary linear discriminant classifiers^14,40,41^ to decode stimulus categories (Go vs No-Go) with 100 ms binning. In brief, for each recording sessions, population vectors were constructed at each time bin and trained with equal number of random Go and No-Go trials (VtGo,VtNo−Go). Then the population decoding vector is given by Wt=VtGo−VtNo−GoWhere VtGo,VtNo−Go are trial-averaged population vectors for Go and No-Go categories respectively. The threshold is defined by bt=−(WtxVtGo+WtxVtNo−Go)2 For each test trial population vector V ^test^_t_ , the linear discriminant function is given by Yt=WtxVtGo+btThe test trial Vttest is assigned to the Go category if Y _t_ ≥0 and to the No-Go category otherwise. The assigned category is compared to the ground truth and the proportion correct defines the accuracy of the classifier. Cross-validation was performed 200 times by randomly choosing train (70% of data) and test trials (30% of data).

Random classifier performance was obtained with 200 label shuffling permutations (the lower bound for the p-value being 1/200 = 0.005) and the comparison was statistically evaluated through permutation tests, comparing the actual performance with the chance-level distribution. Unless mentioned otherwise, all analysis were done using only correct trials (correct rejection No-Go trials and hit Go trials). Temporal evolution of the decoder was computed as the correlation between decoding weight vectors at one time bin against others and time points below chance correlation values were shown as grey in Figure 2b. We also projected the trial averaged activity of test trials onto the unit decoders trained at sound and delay period as shown in Figure 2—figure supplement 4.”

The code is now available in the following repository: https://github.com/rupeshjnu/A1-Category

I would have liked the information about the population size to be in the Results section rather than only buried deep in the methods; the populations themselves are really quite small (mean 5 / 11 neurons in the two animals) which is useful in interpreting the modest performance of the decoder (which is clearly above chance but not that much so). Also how confident are they that all units are in A1 as the array sounds like it's quite large (potentially larger than A1) to me?

We agree that it is best to make the information readily available to the reader in the Results section. We added the following comment regarding the single-session decoding in the Results:

“Note that decoding was performed at the single-session level (11.3±4.9 neurons per session for ferret P; ±std; n = 35 sessions and 5.2±3.2 neurons per session for ferret T; n = 39 sessions), explaining the modest but above chance-level performance of the decoder.“

The array size is relatively large (2.8 x 1.2 mm, with 8x4 electrodes and 0.4 μm distance between the electrodes) but perfectly fits within ferret A1 which is about 4-5 mm-long and 3-4 mm-wide (Bizley et al., 2005. *Cerebral Cortex*; Elgueda et al., 2019. *Nature Neuro*). We now detail the procedure for making sure we were in A1 in the methods:

“Using a stereotaxic apparatus, the headpost was mounted on the skull using methyl methacrylate based dental adhesive resin cement. Stainless steel screws were anchored along the areas surrounding the auditory cortex leaving a cavity for easy access to the auditory cortex. The typical horseshoe shape of the auditory cortex was marked with nail polish. Finally the surrounding areas are filled with poly-methyl methacrylate (PMMA) based bone cement to stabilize the implant. […] Under surgical anesthesia (isoflurane 1 %), we removed the cement above the location marked during the surgery. We then performed a 4 mm x 4 mm craniotomy. This craniotomy allowed us to identify the core regions of the auditory cortex (middle ectosylvian gyrus) by visual inspection of the tip of the ectosylvian gyrus. We carefully removed the transparent dura to ensure that the array penetrated the brain without additional strain. […] Primary auditory cortical responses were identified by analyzing tuning properties to 100 ms tone pips of random frequencies spanning 4 octaves and temporally orthogonal ripple combinations (STRF)^39^. A1 responses show sharp tuning to random tones and single peak, short latency STRFs^5,10,21^.“

There are many places in the manuscript where it's not obvious whether the data is from one animal or both (one assumes one animal, as the figures list only a single contingency for high/low rates). The data for both animals are very clearly laid out in the supplemental material but not always well described in the main manuscript.

Most of the results in the main figures are derived from one animal such as Figure 1a-1d, Figure 2, Figure 3, Figure 4a,c,d and Figure 5a-5b. The validity of these results in the second animal is shown in corresponding supplementary figures. We now mention neuron and session numbers from either one or two ferrets in each figure caption.

Reviewer #3 (Recommendations for the authors):This work investigated the activity of neurons from the primary auditory cortex (A1) of ferrets performing a click–rate categorization go/no–go task or passively listening to these sounds. The authors found that the population of recorded A1 neurons shows a different firing pattern for go vs. no–go stimuli, not only during the stimulus presentation but also during a delay period before the licking response. Prediction of the go vs. no–go categories via neural decoding analysis revealed that these categories were decodable during both the stimulus and delay periods, but the population code was different between these two periods.The authors provide clear evidence of differences in neural activity patterns for correct trials with go vs no–go stimuli. However, it is not completely clear that these observations reflect auditory categorization as the authors suggest. Most of the data presented seems consistent with alternative interpretations such as a representation of expected reward or pre–motor signals in the auditory cortex. For example: (1) the differences in neural activity between go and no–go stimuli are not present during the passive presentation (Figure 1f) when animals are presumably not licking or expecting reward; (2) the dynamics of neural activity changes consistently with movement when comparing (invalid) early licks, (invalid) late licks and (valid) hit trials (Figure 4c); and (3) the population code that enables decoding of go vs no–go stimuli changes between the stimulus presentation period and the delay period, which suggests a change in what is being represented during these periods (which could be mostly stimulus identity in the first period and motor–preparation signals in the second period). As such, the claim that neural activity reflects the categorization of the stimuli rather than the representation of other variables does not seem fully supported.The authors try to address some of these concerns in the discussion by suggesting that motor–related activity is expected to have a short latency (~100 ms). However, from their experiments, it seems difficult to rule out that signals related to motor preparation or reward expectation (at possibly multiple latencies) are the main drivers of the observed effects.

We thank the reviewer for the comments and observations on our results. We are totally open to all these interpretations for two main reasons. First, we have observed categorical tuning in the population average, both during passive and task-engaged states. Then, the emergence of early categorical signals and the task-induced suppression of the No-Go sounds at the population-level are independent of the licking activity. Second, we believe that the sustained delay activity whose dynamics scales with the licking action is interesting irrespective of its origin. But we fully agree with the reviewer that the sustained activity we observed could be explained by other different factors. For this reason, we toned down the description of this activity in the abstract and introduction:

“The population code underwent an abrupt change at stimulus offset, with sustained responses after the Go sounds during the delay period.” And

“Third, at stimulus offset, the population code changed abruptly and a large fraction of neurons maintained sustained responses after Go sounds throughout the delay epoch.”

We also take care to mention this point in the results:

“We consider further the possible link of this delay activity pattern with reward expectation or motor preparation (see Discussion). For practical reasons, we will refer to the delay activity as category-related in the following sections.“

In addition, we now discuss several alternative interpretations of this activity in the Discussion: categorical activity tied to choice, reward expectation and motor preparation:

“We note that this sustained activity after Go sounds can be caused by at least three different factors.

First, delay activity in A1 could be the result of a feedback signal from higher-order decisionrelated areas signaling the chosen category and maintaining it in memory by engaging A1 in a network of parietal and frontal areas^33^. In this framework, previous works in the somatosensory system have shown that choice-related information flows bidirectionally between primary and secondary somatosensory cortices, with choice-related information emerging in primary sensory cortex, then fed to downstream areas that further feedback enriched choice-related information to the primary field^34^. Consistent with this hypothesis, we have demonstrated that trial-to-trial fluctuations of categorical responses during the sound period correlate with the amplitude of the delay categorical signal (Figure 4b), possibly suggesting that category-related information during the stimulus and delay periods are part of this communication loop.

A second interpretation is that delay activity is the result of motor preparation that would unfold over several hundreds of milliseconds^16^. The pattern of activity observed during early trials is consistent with this view, with faster build-up when the animals licked earlier. However, a similar pattern was not observed on false alarm trials, contrary to what would be predicted for this interpretation.

A last possibility is that delay activity signals reward expectation^35^, a type of response which increases through learning^16,36^. This delay activity would then reflect post-learning residual topdown feedback, or alternatively may be the signature of an eligibility trace, i.e. a persistent activity necessary for bridging the gap between the sound and the response window^37^. Overall, disentangling these different interpretations will require further experiments.”

If we define perceptual categorization as a maximization of perceptual differences between categories and a minimization of the differences within a category, investigating the neural representation of auditory categories may require a more nuanced comparison of how well one can decode stimuli within vs. across categories from neural activity.

We agree with the reviewer that we did not sufficiently stress this point in the original version, although we used a category index (CI) as a direct measure of the difference in discriminability for pairs of stimuli that were within or across categories. This index compares the distances between projections of stimuli onto the categorical neural axis (Figure 3figure supplement 1a,b). We changed the text to highlight how the CI relates to grouping and separability between stimulus pairs:

“Categorization can be defined as the maximization of neural differences between categories and a minimization of the differences within a category. Therefore we investigated stimulus discriminability along the category neural axis. We designed a category index (see Methods) which compared the neural distance between stimuli at the category boundary (12 and 16 Hz) against pairs of stimuli within categories. We found that categories were effectively present in both passive and task-engaged states during stimulus presentation with equal magnitude (Figure 3a,b and Figure 3—figure supplement 1a,b). In contrast, significant categories were found only in the task-engaged state during the delay period (Figure 3c,d).“

Additionally, we believe that we did address this question from two different perspectives in the manuscript. First, we showed the neural distances between pairs of stimuli projected on the category decoder (Figure 2—figure supplement 4). The grouping of stimuli within the same category can be seen during the stimulus period, and even more during the delay (Figure 2figure supplement 4d) and the response window (Figure 2—figure supplement 4f).

– The manuscript would benefit from a discussion of alternative explanations related to reward expectation.

Thank you for the suggestion; we review these alternative explanations in the discussion now (see comment 2 pages above).

– The differences in neural activity seem compelling, but the author may want to de–emphasize the idea that these changes are associated with a neural representation of auditory categories.

Similarly, we toned down this point throughout the manuscript. For instance, in the abstract:

“Third, at stimulus offset, the population code changed abruptly and a large fraction of neurons maintained sustained responses after Go sounds throughout the delay epoch.”

or in the discussion:

“Categorical responses changed at stimulus offset to maintain a Go-specific prolonged activity during the delay period, consistent with other studies showing choice-related activity in A1^25–28^. We do not interpret this activity as efferent copies directly sent from motor regions^29^, as one would expect such motor-related activity to have shorter latencies (~100-300 ms^14,30^). Here, sound category was decodable throughout the entire delay period (Figure 2a), which does not match with short-latency efference copies. We have also found that the delay activity is not closely locked in time to licks since false alarm trials, in which the animals licked during the response window, were not decoded as hit trials during the delay period (Figure 5b).

Other studies investigating the role of A1 in auditory Go/No-Go tasks with a delay did not analyze neuronal responses after Go sounds^31,32^, which may explain why this sustained activity has not been previously reported. Interestingly, delay activity in sensory cortices is thought not to be causally involved in the behavioral response of trained animals (causal inactivations in V1^33^ and A1^31^). We note that this sustained activity after Go sounds can be caused by at least three different factors.”

– Figure 1e: because the structure can appear from random data when sorted, a supplementary figure showing neurons sorted by the delay during passive would illustrate that effects shown in this figure are not just the result of sorting for the active condition.

The increase in delay period activity during the active context is not due to sorting. We added a new supplementary figure (Figure 1—figure supplement 3) by sorting using passive state which shows the sorting done from the passive data has no effect on the claim mentioned in Figure 1d:

“Ranking neurons by delay activity during the passive condition did not reveal a similar pattern neither in passive or active states (Figure 1—figure supplement 3), reflecting a pattern of response specific to the task-engaged delay period.”

– It would be useful to clarify whether the "increase" associated with Figure 1f (during the delay period) is with respect to the spontaneous or sound–evoked activity.

Figure 1f modulation index is with respect to the spontaneous activity. This is now mentioned in the caption and in the figure.

– Clarify what "R.W" means. I don't think that is a standard acronym.

The R.W. in all figures corresponds to the response window. We now mention it in the figure captions.

– Figure 1f: specify what period(s) of activity the modulation index refers to.

The time-period used to calculate the modulation index in Figure 1f is mentioned in the caption.

– The authors need to clarify whether the animals are head–fixed or freely moving during training and recordings. While they mentioned "To obtain stable neurophysiological recordings we implanted the ferrets with a stainless steel headpost", it's not clear when the headpost was used since the electrodes were chronically implanted.

The ferrets were head-fixed during the training and recording sessions. We now mention it in the methods:

“Each recording session consisted of passive and active sessions. Recordings were performed head-fixed in a soundproof chamber. In the passive sessions the water spout was removed.“

– Authors should also specify how sounds were delivered (Figure 1 seems to indicate the ferrets had headphones).

Thank you for the observation. We have clarified the details of the sound presentation in the methods section:

“The animals were head fixed in a custom-made tube during training and recording sessions and the stimuli were presented from a calibrated earphone (Sennheiser IE800, HDVA 600 amplifier). Ferrets had to classify click trains into two categories: target (Go) and non-target (NoGo) depending on the rates of click trains. Six rates were used, from 4 to 24 Hz in 4 Hz steps, and with a category boundary fixed at 14 Hz. To ensure the dissociation between categories and stimulus rates, one animal was trained with low rates as the Go sounds, while the second animal classified high rates as the Go sounds.“

– Authors should be clearer about the passive stimulation sessions. Are the animals licking? are there other differences compared to the active sessions (e.g., inter–trial interval)?

The passive sessions were identical to active sessions in terms of stimulus presentation and duration, with the only difference being the water spout was absent. There was no change of inter-trial interval. As far as the licking is considered, the animal did not show any licking behavior since the passive sessions were always first to record and the animal readily recognized the absence of water spout. After the passive session, we brought the waterspout back in place and provided some water drops before starting the active session recording, so that animals recognized the presence of the spout.

– The first mention of "the naive animal" comes out of nowhere. The authors should introduce that there is a naive animal used for control experiments.

Thank you for the observation. We now added a sentence to introduce the naive animal in the result section:

“As a control, we performed recordings in an untrained (naive) animal using the same set of stimuli (Figure 2—figure supplement 2a). The accuracy of passive context decoder was also similar to what is observed in the naive animal (Figure 2—figure supplement 2b).”

– Figure 2 caption: I don't know what "resp." means.

The caption is now corrected.

– Figure 4C: y–label should say "categorical".

We have corrected this typo.

– Figure 5a: the caption says "cyan" but it looks purple to me.

We have corrected the caption to purple.